# Interpretability of Language Models for Learning Hierarchical Structures

## Abstract

Transformer-based language models are effective but complex, and understanding their inner workings is a significant challenge. Previous research has primarily explored how these models handle simple tasks like name copying or selection, and we extend this by investigating how these models grasp complex, recursive language structures defined by context-free grammars (CFGs). We introduce a family of synthetic CFGs that produce hierarchical rules, capable of generating lengthy sentences (e.g., hundreds of tokens) that are locally ambiguous and require dynamic programming to parse. Despite this complexity, we demonstrate that generative models like GPT can accurately learn this CFG language and generate sentences based on it. We explore the model's internals, revealing that its hidden states precisely capture the structure of CFGs, and its attention patterns resemble the information passing in a dynamic programming algorithm.

## 1 Introduction

Transformer-based language models, like GPT (OpenAI, 2023), are powerful but mysterious; many studies attempt to uncover the inner workings of transformers. Perhaps the simplest observation is that attention heads can pair closing brackets with open ones, see the concurrent work and the references therein (Zhang et al., 2023). Others also demonstrate that transformer can store key-value knowledge pairs by storing value in the hidden embedding of keys (see Allen-Zhu & Li (2023) and the references therein).

The seminal work from Anthropic (Elhage et al., 2021; Olsson et al., 2022) focuses on *induction heads*, which are logic operations *on the input level* (such as [A][B]...[A] implies the next token should be [B]). They "hypothesized" that induction heads may exist to "match and copy more abstract and sophisticated linguistic features, rather than precise tokens", yet they acknowledge that they "don't have a strong framework for mechanistically understanding" this.

The *interpretability in the wild* paper (Wang et al., 2022) explored many different types of attention heads, including "copy head", "name mover head", "inhibition head", etc. Most notably, they explained how GPT2 predicts the next token "Mary" given prefix "When Mary and John went to the store, John gave a drink to [...]" This requires some logical reasoning by selecting (not naively copying) what is the right name. While this result is very inspiring, there exists very simple rule-based algorithm to achieve the same.

In practice, transformers perform much more complex operations, yet, there is an inherent difficulty in interpreting those models: *To interpret how transformer performs a certain task, there must be a well-defined algorithm to solve it so one can argue that the inner representations of the transformer align with the algorithm.* Almost all of the "impressive skills" demonstrated by state-of-the-art language models are beyond solvable by any other known algorithm. Motivated by these, we ask: *Is there a setting for us to understand **how** language models perform **hard** tasks, involving deep logics / reasoning / computation chains?*

We propose to tackle this question in a *controlled* setting where the languages are generated *synthetically* using context-free grammars (CFGs). CFGs, which include terminal (T) and nonterminal (NT) symbols, a root symbol, and production rules, can *hierarchically* produce highly structured expressions. A string is part of CFG language if a rule sequence can transform the root symbol into this string, and the language model is asked to complete the given partial strings from the CFG. We pick CFG because, there exists textbook-level, yet quite difficult dynamic programming (DP)

```
root |->20 21       19|->18 16 18   16|->15 15      13|->11 12     10|->8 9 9   7|->2 2 1
root |->20 19 21    19|->17 18      16|->13 15 13   13|->12 11 12  10|->9 7 9   7|->3 2 2    an example sentence
root |->21 19 19    19|->18 18      16|->14 13      13|->10 12 11  10|->7 9 9   7|->3 1 2
root |->20 20       20|->16 16      16|->14 14      14|->10 12     11|->8 8     7|->3 2      332213123312113123211322312311211121132113223113111
                    20|->16 17      17|->15 14 13   14|->12 10 12  11|->9 7     8|->3 1 1    322333123121112131133112132121313333123221213232
                    20|->17 16 18   17|->14 15      14|->12 11     11|->9 7 7   8|->1 2      221111213322131311311311311111132312331331333113331
                    21|->18 17      17|->15 14      14|->10 12 12  12|->7 9 7   8|->3 3 1    333332231211311112122111112112333123311211113313333
                    21|->17 16      18|->14 15 13   15|->10 11 11  12|->9 8     9|->1 2 1    331123333131111333312113211321131212111333332121111121
                    21|->16 17 18   18|->15 13 13   15|->11 11 10  12|->8 8 9   9|->3 3      213222232231332211132211323232313111213223223221
                    21|->16 18      18|->13 15      15|->10 10                 9|->1 1      211133331121322221332211212133121331331332212213221
                                                    15|->12 12 11                           211121333123233312
```

Figure 1: An example CFG used in our experiments. It generates long (e.g., *length 354* in this example) and ambiguous strings. Determining if a string $x$ belongs to the CFG typically requires dynamic programming, even when the CFG rules are known.

algorithm to solve CFG instances.[1] Generally,

- We wish to capture *long-range* dependencies via CFG. The simplest example is bracket matching, in `...Y(...)[[...]{...}]{...}X`, the next symbol X could depend on Y that was hundreds of tokens before. Another example is coding, where `goto jumpback` can only be used if `jumpback` is a valid line number that could be hundreds of lines ago.
- We wish to capture *local ambiguity*. A coding grammar (like python) can be parsed using greedy without ambiguity, so does bracket matching — once locally seen `...()...` we know the two parentheses must be paired together. We *focus on hard CFGs* that require global planning via *dynamic programming* to parse.

Most popular choices of CFGs do not satisfy the two above properties. Notably, the English CFG (e.g., derived from Penn TreeBank) has an average length of 28 tokens (too short), and is not very locally ambiguous (e.g., `RB JJ` or `JJ PP` imply their parent must be `ADJP`). As we show in Appendix I, such CFGs can even be learned using tiny GPT2 models with $\sim$ 100k parameters. Thus, *it is too easy* for our interpretability purpose.

**For such reason, we design our own synthetic CFG languages.** We give one example in Figure 1 and discuss a family of 7 such CFGs with varying difficulties in Section 2 (we have 15 more in the appendix).[2] We *pre-train* GPT-2 (Radford et al., 2019), denoted by GPT for short, on a language modeling task using a large corpus of strings sampled from our constructed CFGs. We test the model's accuracy and diversity by feeding it prefixes from the CFG (or no prefix, just the starting token) and observing if it can generate accurate completions.

It is perhaps evident from Figure 1 that *even if* the CFG tree is given, *deciding* if the string belongs to this language for a real person may require a scratch paper and perhaps half an hour, not to say to learn such CFG from scratch. However, we demonstrate that GPT can learn such CFGs, and using rotary or relative attentions is crucial, especially for complex CFGs (**Results 1-3**). Additionally, we examine attention patterns and hidden states to understand how GPT achieves this. Specifically, we:

- **Results 4-5.** Develop a multi-head linear probing method to verify that the model's hidden states linearly encode NT information almost perfectly, a significant finding as pre-training does not expose the CFG structure. (In contrast, encoder models like BERT do not.)
- **Results 6-9.** Introduce methods to visualize and quantify attention patterns, demonstrating that GPT learns position-based and boundary-based attentions, contributing to understanding how it learns CFG's regularity, periodicity, and hierarchical structure.
- **Corollary.** Suggest that GPT models learn CFGs by *implementing a dynamic programming-like algorithm*. The boundary-based attention allows a token to attend to its closest NT symbols in the CFG tree, even when separated by hundreds of tokens. This resembles DP, in which the CFG parsing on a sequence $1...i$ needs to be "concatenated" with another sequence $i+1...j$ in order to form a solution to a larger problem on $1...j$. See Figure 2+8 for illustrations.

In Appendix B, we also explore *implicit CFGs* (Post & Bergsma, 2013), where each T symbol is a bag of words, and show that GPT simply learns to encode the word information on its embedding layer. We also investigate *model robustness* using CFGs, showcasing *under what conditions* the model can auto-correct errors and generate valid CFGs from a corrupted prefix (e.g., randomly flipping 15% of the symbols in the prefix). These results are numbered 10 through 13.

---

[1] Not to say in the theory community, CFGs are also used to model some rich, recursive structure in languages, including some logics, grammars, formats, expressions, patterns, etc.

[2] A benefit of using synthetic data is to control the difficulty of the data, so that we can observe how transformers learn to solve tasks at different difficulty levels, and observe their difference.

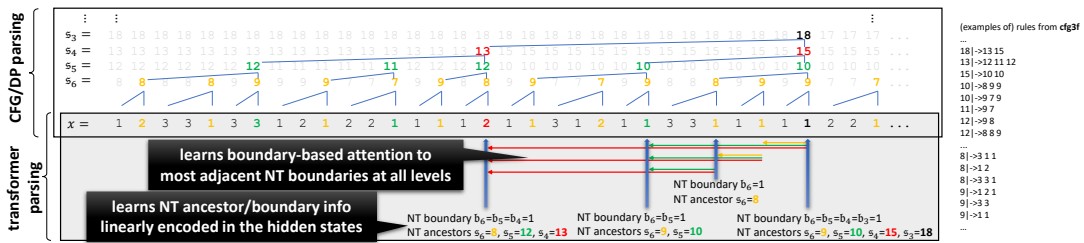

Figure 2: An example string $x$ from $\mathcal{G} = \text{cfg3f}$. Though formally defined in Section 2, bold symbols in color represent *NT boundaries* which mark the ending positions of the parsed CFG subtrees at various levels $\ell$: we denote by $\mathfrak{b}_\ell(i) = 1$ if position $i$ is at the NT boundary for level $\ell$. The *NT ancestor* $\mathfrak{s}_\ell(i)$ represents the tree node's name at level $\ell$ for a symbol at position $i$.

## 2 OUR SYNTHETIC CONTEXT-FREE GRAMMARS

A probabilistic context-free grammar (CFG) is a formal system defining a string distribution using production rules. It comprises four components: terminal symbols ($\mathbf{T}$), nonterminal symbols ($\mathbf{NT}$), a root symbol ($root \in \mathbf{NT}$), and production rules ($\mathcal{R}$). We represent a CFG as $\mathcal{G} = (\mathbf{T}, \mathbf{NT}, \mathcal{R})$, with $L(\mathcal{G})$ denoting the string distribution generated by $\mathcal{G}$.

We mostly focus on $L$-level CFGs where each level $\ell \in [L]$ corresponds to a set of symbols $\mathbf{NT}_\ell$ with $\mathbf{NT}_\ell \subseteq \mathbf{NT}$ for $\ell < L$, $\mathbf{NT}_L = \mathbf{T}$, and $\mathbf{NT}_1 = \{root\}$. Symbols at different levels are disjoint: $\mathbf{NT}_i \cap \mathbf{NT}_j = \varnothing$ for $i \neq j$. We consider rules of length 2 or 3, denoted as $\mathcal{R} = (\mathcal{R}_1, \ldots, \mathcal{R}_{L-1})$, where each $\mathcal{R}_\ell$ consists of rules in the form:

$$r = (a \mapsto b, c, d) \quad \text{or} \quad r = (a \mapsto b, c) \quad \text{for} \quad a \in \mathbf{NT}_\ell \quad \text{and} \quad b, c, d \in \mathbf{NT}_{\ell+1}$$

Given a non-terminal symbol $a \in \mathbf{NT}$ and any rule $r = (a \mapsto \star)$, we say $a \in r$. For each $a \in \mathbf{NT}$, its associated set of rules is $\mathcal{R}(a) := \{r \mid r \in \mathcal{R}_\ell \wedge a \in r\}$, its *degree* is $|\mathcal{R}(a)|$, and the CFG's *size* is $(|\mathbf{NT}_1|, |\mathbf{NT}_2|, \ldots, |\mathbf{NT}_L|)$.

**Generating from CFG.** To generate samples $x$ from $L(\mathcal{G})$, follow these steps:

1. Start with the $root$ symbol $\mathbf{NT}_1$.
2. For each layer $\ell < L$, keep a sequence of symbols $s_\ell = (s_{\ell,1}, \cdots, s_{\ell,m_\ell})$.
3. For the next layer, randomly sample a rule $r \in \mathcal{R}(s_{\ell,i})$ for each $s_{\ell,i}$ with uniform probability.[3] Replace $s_{\ell,i}$ with $b, c, d$ if $r = (s_{\ell,i} \mapsto b, c, d)$, or with $b, c$ if $r = (s_{\ell,i} \mapsto b, c)$. Let the resulting sequence be $s_\ell = (s_{\ell+1,1}, \cdots, s_{\ell+1,m_{\ell+1}})$.
4. During generation, when a rule $s_{\ell,i} \mapsto s_{\ell+1,j}, s_{\ell+1,j+1}$ is applied, define the parent $\text{par}_{\ell+1}(j) = \text{par}_{\ell+1}(j+1) := i$ (and similarly if the rule of $s_{\ell,i}$ is of length 3).
5. Define **NT ancestor indices** $\mathfrak{p} = (\mathfrak{p}_1(i), \ldots, \mathfrak{p}_L(i))_{i \in [m_L]}$ and **NT ancestor symbols** $\mathfrak{s} = (\mathfrak{s}_1(i), \ldots, \mathfrak{s}_L(i))_{i \in [m_L]}$ as shown in Figure 2:

$$\mathfrak{p}_L(j) := j \ , \quad \mathfrak{p}_\ell(j) := \text{par}_{\ell+1}(\mathfrak{p}_{\ell+1}(j)) \quad \text{and} \quad \mathfrak{s}_\ell(j) := s_{\ell,\mathfrak{p}_\ell(j)}$$

The final string is $x = s_L = (s_{L,1}, \cdots, s_{L,m_L})$ with $x_i = s_{L,i}$ and length $\mathbf{len}(x) = m_L$. We use $(x, \mathfrak{p}, \mathfrak{s}) \sim L(\mathcal{G})$ to represent $x$ with its associated NT ancestor indices and symbols, sampled according to the generation process. We write $x \sim L(\mathcal{G})$ when $\mathfrak{p}$ and $\mathfrak{s}$ are evident from the context.

**Definition 2.1.** *A symbol $x_i$ in a sample $(x, \mathfrak{p}, \mathfrak{s}) \sim L(\mathcal{G})$ is the **NT boundary / NT end** at level $\ell \in [L-1]$ if $\mathfrak{p}_\ell(i) \neq \mathfrak{p}_\ell(i+1)$ or $i = \mathbf{len}(x)$. We denote $\mathfrak{b}_\ell(i) := \mathbb{1}_{x_i \text{ is the NT boundary at level } \ell}$ as the* **NT-end boundary** *indicator function. The **deepest NT-end** of $i$ is*

$$\mathfrak{b}^\sharp(i) = \min_{\ell \in \{2,3,\ldots,L-1\}} \{\mathfrak{b}_\ell(i) = 1\} \quad \text{or} \perp \text{if set is empty} \ .$$

**The cfg3 synthetic CFG family.** We focus on seven synthetic CFGs of depth $L = 7$ detailed in Section C.1. The hard datasets cfg3b, cfg3i, cfg3h, cfg3g, cfg3f have sizes $(1, 3, 3, 3, 3, 3, 3)$ and increasing difficulties cfg3b $<$ cfg3i $<$ cfg3h $<$ cfg3g $<$ cfg3f. The easy datasets cfg3e1 and

---

[3]For simplicity, we consider the uniform case, eliminating rules with extremely low probability. Such rules complicate the learning of the CFG and the investigation of a transformer's inner workings (e.g., require larger networks and longer training time). Our results do extend to non-uniform cases when the distributions are not heavily unbalanced.

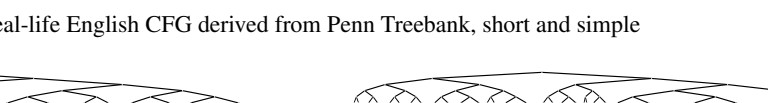

(a) real-life English CFG derived from Penn Treebank, short and simple

(b) a family of max-depth 11 CFGs where rules have length 1 or 2 that GPT can learn, see cfg0 in Appendix I

Figure 3: CFG visual comparisons: *left* is a medium-length sample, and *right* is a 80%-percentile-length sample

cfg3e2 have sizes $(1, 3, 9, 27, 81, 27, 9)$ and $(1, 3, 9, 27, 27, 9, 4)$ respectively. The sequences generated by these CFGs are up to $3^6 = 729$ in length. Typically, the learning difficulty of CFGs *inversely scales* with the number of NT/T symbols, assuming other factors remain constant, because having more NT/T symbols makes the language less ambiguous and more easily parsed using greedy (see Figure 4 and we discuss more in Appendix I). We thus primarily focus on cfg3b, cfg3i, cfg3h, cfg3g, cfg3f.

**Why Such CFGs.** We use CFG as a proxy to study some rich, recursive structure in languages, which can cover some logics, grammars, formats, expressions, patterns, etc. Those structures are diverse yet strict (for example, Chapter 3.1 should be only followed by Chapter 3.1.1, Chapter 4 or Chapter 3.2, not others). The CFGs we consider are non-trivial, with likely over $2^{270} > 10^{80}$ strings in cfg3f among a total of over $3^{300} > 10^{140}$ possible strings of length 300 or more (see the entropy estimation in Figure 4). In particular, Figure 30 in the appendix shows that cfg3f cannot be learned by transformers (much) smaller than GPT2-small. In contrast, the English CFG (e.g., derived from Penn TreeBank) can be learned to good accuracy using tiny GPT2 models with $\sim 100$k parameters — so *it is too easy* for our interpretability purpose.

To obtain the cleanest interpretability result, we have selected a CFG family with a "canonical representation" (e.g., a layered CFG). This *controlled* design choice allows us to demonstrate a strong correlation between the CFG representation and the hidden states in the learned transformer. We also create additional CFG families to examine "not-so-canonical" CFG trees, with results deferred to Appendix I (see an example in Figure 3). *We do not claim* our results encompass all CFGs; our chosen CFGs are already quite challenging for a transformer to learn and can lead to clean hierarchical interpretability results.

## 3 RESULTS 1-3: TRANSFORMER CAN LEARN SUCH CFGS

In this section, we generate a large corpus $\{x^{(i)}\}_{i \in [N]}$ from a synthetic CFG language $L(\mathcal{G})$ in Section 2, and pretrain a (generative, decoder-only) transformer model $F$ on this corpus, treating each terminal symbol as a separate token, using an auto-regressive task (see Appendix C.3 for details). We then evaluate how well the model learns such $L(\mathcal{G})$.

**Models.** We denote the GPT2 small architecture (12-layer, 12-head, 768-dimensions) as GPT (Radford et al., 2019) and implemented its two modern variants. We denote GPT with relative positional attention (He et al., 2020) as $\text{GPT}_{\text{rel}}$, and GPT with rotary positional embedding (Su et al., 2021; Black et al., 2022) as $\text{GPT}_{\text{rot}}$. For purposes in later sections, we introduce two weaker variants. $\text{GPT}_{\text{pos}}$ replaces the attention matrix with a matrix based solely on tokens' relative positions, while $\text{GPT}_{\text{uni}}$ uses a constant, uniform average of past tokens from various window lengths as the attention matrix. Detailed explanations of these variants are in Section C.2.

We quickly summarize our findings and then elaborate them in details.

**Result 1-3** (Figure 4). *The GPT models can effectively learn our synthetic CFGs. Given any prefix, they can generate completion strings*

- *that can perfectly adhere to the CFG rules most of the time,* **(accuracy)**

- *that are sufficiently diverse in the CFG language, and* **(diversity)**

- *that closely follow the probabilistic distribution of the CFG language.* **(probability)**

*Moreover, one had better use rotary or relative attentions; the original* GPT *(with absolute positional embedding) performs even worse than* $\text{GPT}_{\text{uni}}$ *(with uniform attention).*

Figure 4: Generation accuracy (left), entropy (middle), KL-divergence (right) across multiple CFG datasets.
**Observations:** Less ambiguous CFGs (cfg3e1, cfg3e2, as they have fewer NT/T symbols) are easier to learn. Transformers using relative positional embedding (GPT_rel or GPT_pos) are better for learning harder CFGs. The vanilla GPT is worse than even GPT_uni, which is GPT with fixed, uniform attentions.

**Result 1: Completion accuracy.** We evaluate $F$ by letting it generate completions for prefixes $x_{:c} = (x_1, x_2, \cdots, x_c)$ from strings $x$ freshly sampled from $L(\mathcal{G})$. The *generation accuracy* is measured as $\mathbf{Pr}_{x \sim L(G) + \text{randomness of } F}[(x_{:c}, F(x_{:c})) \in L(\mathcal{G})]$. We use multinomial sampling without beam search for generation.[4] Figure 4 (left) shows the generation accuracies for cuts $c = 0$ and $c = 50$. The $c = 0$ result tests the transformer's ability to generate a sentence in the CFG, while $c = 50$ tests its ability to complete a sentence.[5] The results show that the pretrained GPT models can often generate strings that perfectly adhere to the CFG rules for the cfg3 data family.

**Result 2: Generation diversity.** Could it be possible that the pretrained GPT models only memorized a small subset of strings from the CFG? We evaluate this by measuring the diversity of its generated strings. High diversity suggests a better understanding of the CFG rules.

We consider two methods to estimate diversity. One is to estimate the distribution's entropy, which provides a rough estimate of (the $\log_2$ of) the support size, see the middle of Figure 4. The other is to use birthday paradox to theoretically lower bound the support size (Arora & Zhang, 2017). This allows us to make precise claims, such as in the cfg3f dataset, there are at least $4 \times 10^8$ distinct sentential forms derivable from a symbol at levels 1 to 5 or levels 2 to 6; not to say from the root to level 7. Details are in Appendix D. Our general conclusion is that the pre-trained model **does not rely on simply memorizing** a small set of patterns to achieve high completion accuracy.

**Result 3: Distribution comparison.** To fully learn a CFG, it is crucial to learn the distribution of generating probabilities. One naive approach is to compare the marginal distributions $p(a, i)$, for the probability of symbol $a \in \mathbf{NT}_\ell$ appearing at position $i$. We observe a strong alignment between the generation probabilities and the ground-truth, included in Appendix D.2. Another approach is to compute the KL-divergence between the per-symbol conditional distributions. Let $p^*$ be the distribution over strings in the true CFG and $p$ be that from the generative transformer model. Let $S = \{x^{(i)}\}_{i \in [M]}$ be samples from the true CFG distribution. Then, the KL-divergence can be estimated as follows:[6]

$$\frac{1}{|S|} \sum_{x \in S} \frac{1}{\mathbf{len}(x)+1} \sum_{i \in [\mathbf{len}(x)+1]} \sum_{t \in \mathbf{T} \cup \{\text{eos}\}} \mathbf{Pr}_{p^*}[t \mid x_1, \ldots, x_{i-1}] \log \frac{\mathbf{Pr}_{p^*}[t|x_1,\ldots,x_{i-1}]}{\mathbf{Pr}_p[t|x_1,\ldots,x_{i-1}]}$$

In Figure 4 (right) we compare the KL-divergence between the true CFG distribution and the GPT models' output distributions using $M = 20000$ samples.

**Connection to DP.** Result 1-3 (e.g., learning the CFG's marginal distribution) is merely an small step towards showing that the model employs a DP-like approach. Dynamic programming (e.g., the inside-outside algorithm Baker (1979)) can compute marginal distributions of CFGs, and such algorithms can be implemented using nonlinear neural networks like transformers, achieving a global minimum in the auto-regressive training objective.[7] However, the mere existence of a dynamic-programming transformer to obtain the training objective's global minimum is not entirely satisfactory. Does employing an AdamW stochastic optimizer for 100k iterations on the training objective yield such an algorithm? The remainder of this paper will delve deeper to address this question.

---

[4] The last softmax layer converts the model outputs into a probability distribution over (next) symbols. We follow this distribution to generate the next symbol, reflecting the unaltered distribution learned by the transformer. This is the source of the "randomness of $F$" and is often referred to as using "temperature $\tau = 1$."

[5] Our cfg3 family is large enough to ensure a negligible chance of a freshly sampled prefix of length 50 being seen during pretraining.

[6] A nearly identical formula was also used in DuSell & Chiang (2022).

[7] This has been carefully explored for masked language modeling case in Zhao et al. (2023).

Figure 5: After pre-training, hidden states of generative models encode NT-ancestor information. The $NT_\ell$ column represents the accuracy of predicting $\mathfrak{s}_\ell$, the NT ancestors at level $\ell$, via linear probing (4.2).

It also encodes NT boundaries (Appendix E.1); and such information is discovered gradually and *hierarchically* across layers and training epochs (Appendix E.2 and E.3). We compare against a baseline which is the encoding from a randomly-intialized GPT, GPT$_{rand}$ (serving as a neural-tangent kernel baseline). We also compare against DeBERTa, illustrating that BERT-like models are less effective in learning NT information at levels close to the CFG root.

# 4 RESULTS 4-5: HOW DO TRANSFORMERS LEARN CFGS?

In this section, we delve into the learned representation of the transformer to understand *how* it encodes CFGs. We employ various measurements to probe the representation and gain insights.

**Recall classical way to solve CFGs.** Given CFG $\mathcal{G}$, the classical way to verify if a sequence $x$ satisfies $L(\mathcal{G})$ is to use dynamic programming (DP) (Sakai, 1961; Sipser, 2012). One possible implementation of DP involves using the function $\mathsf{DP}(i, j, a)$, which determines whether or not $x_{i+1}, x_{i+1} \ldots, x_j$ can be generated from symbol $a$ following the CFG rules. From this DP representation, a DP recurrent formula can be easily derived.[8]

In the context of this paper, any sequence $x \sim L(\mathcal{G})$ that satisfies the CFG must satisfy the following conditions:

$$\mathfrak{b}_\ell(i) = 1, \mathfrak{b}_\ell(j) = 1, \forall k \in (i, j), \mathfrak{b}_\ell(k) = 0 \text{ and } \mathfrak{s}_\ell(j) = a \implies \mathsf{DP}(i, j, a) = 1 \qquad (4.1)$$

(recall the NT-boundary $\mathfrak{b}_\ell$ and the NT-ancestor $\mathfrak{s}_\ell$ notions from Section 2). Note that (4.1) is not an "if and only if" condition because there may be a subproblem $\mathsf{DP}(i, j, a) = 1$ that does not lie on the final CFG parsing tree but is still locally parsable by some valid CFG subtree. However, (4.1) provides a "backbone" of subproblems, where verifying their $\mathsf{DP}(i, j, a) = 1$ values *certifies* that the sentence $x$ is a valid string from $L(\mathcal{G})$. It is worth mentioning that there are **exponentially many** implementations of the same DP algorithm[9] and **not all** $(i, j, a)$ tuples need to be computed in $\mathsf{DP}(i, j, a)$. Only those in the "backbone" are necessary.

**Connecting to transformer.** In this section, we investigate whether pre-trained transformer $F$ also implicitly encodes the NT ancestor and boundary information. If it does, this suggests that the transformer contains sufficient information to support all the $\mathsf{DP}(i, j, a)$ values in the backbone. This is a significant finding, considering that transformer $F$ is trained solely on the auto-regressive task without any exposure to NT information. If it does encode the NT information after pretraining, it means that the model can both generate and certify sentences in the CFG language.

## 4.1 RESULT 4: TRANSFORMER'S LAST LAYER ENCODES NT ANCESTORS/BOUNDARIES

Let $l$ be the *last layer* of the transformer (other layers are studied in Appendix E.2). Given an input string $x$, we denote the hidden state of the transformer at layer $l$ and position $i$ as $E_i(x) \in \mathbb{R}^d$. We first investigate whether a linear function can predict $\big(\mathfrak{b}_1(i), \ldots, \mathfrak{b}_L(i)\big)_{i \in [\mathbf{len}(x)]}$ and $\big(\mathfrak{s}_1(i), \ldots, \mathfrak{s}_L(i)\big)_{i \in [\mathbf{len}(x)]}$ using the full $\big(E_i(x)\big)_{i \in [\mathbf{len}(x)]}$. If possible, it implies that the last-layer hidden states *encode the CFG's structural information up to a linear transformation*.

---

[8]For example, one can compute $\mathsf{DP}(i, j, a) = 1$ if and only if there exists $i = i_1 < i_2 < \cdots < i_k = j$ such that $\mathsf{DP}(i_r, i_{r+1}, b_r) = 1$ for all $r \in [k-1]$ and $a \to b_1, b_2, \ldots, b_k$ is a rule of the CFG. Implementing this naively would result in a $O(\mathbf{len}^4)$ algorithm for CFGs with a maximum rule length of 3. However, it can be implemented more efficiently with $O(\mathbf{len}^3)$ time by introducing auxiliary nodes (e.g., via binarization).

[9]Each inner loop of the dynamic programming can proceed in any arbitrary order, not limited to $k = i..j$ or $k = j..i$, and the algorithm can prune and break early. This gives a safe estimate of at least $(n!)^{\Omega(n^2)}$ possible implementations. Furthermore, there are at least $2^{\Omega(n)}$ ways to perform binarization, meaning to break length-3 rules to length-2 ones. This is just to detect if a given string of length $n$ belongs to the CFG.

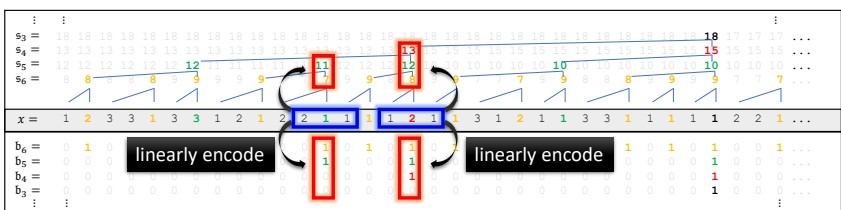

Figure 6: Illustration of Result 5 + Figure 6: GPT's last layer hidden states at the **blue** positions linearly encode the NT ancestor and boundary information in the **red** boxes very well. (They may not encode NT ancestors for smaller levels because that may not be information-theoretically possible.)

**Multi-head linear probing (full).** Due to the high dimensionality of this linear function (e.g., $\mathbf{len}(x) = 300$ and $d = 768$ yield $300 \times 768$ dimensions) and *variable string lengths*, we propose a multi-head linear function for efficient learning. We consider a set of linear functions $f_r : \mathbb{R}^d \to \mathbb{R}^{|\mathbf{NT}|}$, where $r \in [H]$ and $H$ is the number of "heads". To predict any $\mathfrak{s}_\ell(i)$, we apply:

$$G_i(x) = \sum_{r \in [H], k \in [\mathbf{len}(x)]} w_{r, i \to k} \cdot f_r(E_k(x)) \in \mathbb{R}^{|\mathbf{NT}|} \tag{4.2}$$

where $w_{r, i \to k} := \frac{\exp(\langle P_{i,r}, P_{k,r} \rangle)}{\sum_{k' \in [\mathbf{len}(x)]} \exp(\langle P_{i,r}, P_{k',r} \rangle)}$ for trainable parameters $P_{i,r} \in \mathbb{R}^{d'}$. $G_i$ can be seen as a "multi-head attention" over linear functions. We train $G_i(x) \in \mathbb{R}^{|\mathbf{NT}|}$ using the cross-entropy loss to predict $(\mathfrak{s}_\ell(i))_{\ell \in [L]}$. Despite having multiple heads,

$$G_i(x) \text{ **is still a linear function over** } (E_k(x))_{k \in [\mathbf{len}(x)]}$$

as the linear weights $w_{r, i \to k}$ depend only on positions $i$ and $k$, not on $x$. Similarly, we train $G'_i(x) \in \mathbb{R}^L$ using the logistic loss to predict the binary values $(\mathfrak{b}_\ell(i))_{\ell \in [L]}$. Details are in Section C.4.

Using such multi-head linear probing, we discover that:

**Result 4** (Figure 5). *Pre-training allows GPT models to **almost perfectly encode** the NT ancestor $\mathfrak{s}_\ell(i)$ and NT boundary $\mathfrak{b}_\ell(i)$ information in the last transformer layer's hidden states $(E_k(x))_{k \in [\mathbf{len}(x)]}$, up to a **linear** transformation. In contrast, encoder models (like* deBERTa*) may not learn **deep** NT information very well.*[10]

But, do we need this full layer for linear probing? We explore next.

## 4.2 RESULT 5: NT ANCESTORS ARE ENCODED A̲T̲ NT BOUNDARIES

Above, we used the *full* hidden layer, $(E_i(x))_{i \in [\mathbf{len}(x)]}$, to predict $(\mathfrak{s}_\ell(i))_{\ell \in [L]}$ for *each* position $i$. This is essential since it's information-theoretically impossible to extract **all of $i$'s NT ancestors** by only reading $E_i(x)$ or even all hidden states to its *left*, especially if $x_i$ is the start of a string or a subtree in the CFG. But, how about those ones information-theoretically possible? In particular, how about predicting $\mathfrak{s}_\ell(i)$ at locations $i$ with $\mathfrak{b}_\ell(i) = 1$ — i.e., at the end of the CFG subtrees.

**Multi-head linear probing (diagonal).** We consider a neighborhood of position $i$ in the hidden states, say $E_{i \pm 1}(x)$, and use that for linear probing. In symbols, we replace $w_{r, i \to k}$ in (4.2) with zeros for $|i - k| > 1$ (tridiagonal masking), or with zeros for $i \neq k$ (diagonal masking).

$$G_i(x) = \sum_{r \in [H], k \in [\mathbf{len}(x)], |i-k| \leq \delta} w_{r, i \to k} \cdot f_r(E_k(x)) \in \mathbb{R}^{|\mathbf{NT}|} \qquad \text{where } \delta = 0 \text{ or } 1 \tag{4.3}$$

**Result 5** (Figure 6). *For GPT models, the information of position $i$'s NT ancestor/boundary is **locally encoded around position** $i \pm 1$ when $i$ is on the NT boundary. This is because:*

- *At NT boundaries (i.e., $\mathfrak{b}_\ell(x) = 1$), diagonal or tridiagonal multi-head linear probing (4.3) is adequate for accurately predicting the NT ancestors $\mathfrak{s}_\ell(x)$ (see Figure 9 on Page 13).*

---

[10]Among encoder-based models, deBERTa (He et al., 2020) is a modern variant of BERT, which is equipped with relative attentions. It is expected that encoder-based models do not learn very deep NT information, because in a masked-language modeling (MLM) task, the model only needs to figure out the missing token from its surrounding, say, 20 tokens. This can be done by pattern matching, as opposed to a global planning process like dynamic programming.


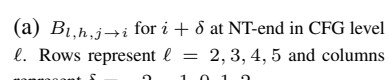
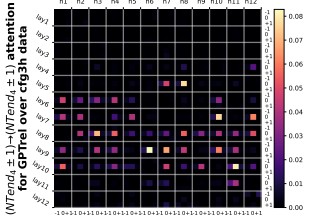
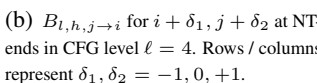
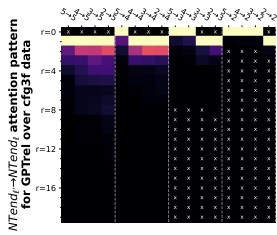

(a) $B_{l,h,j \to i}$ for $i + \delta$ at NT-end in CFG level $\ell$. Rows represent $\ell = 2, 3, 4, 5$ and columns represent $\delta = -2, -1, 0, 1, 2$.

(b) $B_{l,h,j \to i}$ for $i + \delta_1, j + \delta_2$ at NT-ends in CFG level $\ell = 4$. Rows / columns represent $\delta_1, \delta_2 = -1, 0, +1$.

(c) $B_{l,h,\ell' \to \ell,r}^{\text{end} \to \text{end}}$ for NT-ends between CFG levels $\ell' \to \ell$. Rows represent $r$ and columns $\ell' \to \ell$. "×" means empty entries.

Figure 7: After pretrained on our CFG data, GPT model's attention layers have a strong bias towards " NT-end at level $\ell'$ to the most adjacent NT-end at $\ell$ ", for even different $\ell, \ell'$. For definitions see Section 5.2, and more experiments see Appendix F.2, F.3 and F.4. **Corollary:** this is evidence that the model uses dynamic-programming like approach to learn such hard, synthetic CFGs (see discussions in Section 5.3).

- *Such masking is also sufficient for accurately predicting NT boundaries $\mathfrak{b}_\ell(i)$ (deferred to Figure 18 in Appendix E.1).*

*In contrast, encoder models like* `deBERTa` *do* not *store deep NT information at the NT boundaries.*

**Related work.** Our probing approach is akin to the seminal work by Hewitt & Manning (2019), which uses linear probing to examine the correlation between BERT's hidden states and the parse tree distance metric (similar to NT-distance in our language). Subsequent studies (Shi et al., 2022; Zhao et al., 2023; Maudslay & Cotterell, 2021; Manning et al., 2020; Vilares et al., 2020; Wu et al., 2020; Arps et al., 2022) have explored various probing techniques to suggest that BERT-like transformers can approximate CFGs from natural languages.

Our approach differs in that we use synthetic data to demonstrate that linear probing can *almost perfectly* recover NT ancestors and boundaries, even for complex CFGs that generate strings exceeding hundreds of tokens. We focus on pre-training *generative (decoder-only)* language models. For a non-generative, encoder-based model like BERT (Kenton & Toutanova, 2019) or its modern variant `deBERTa` (He et al., 2020), they do not learn *deep* (i.e., close to the CFG root) NT information very well, as shown in Result 4-5.

Our results, along with Section 5, provide evidence that generative language models like GPT-2 employ a dynamic-programming-like approach to generate CFGs, while encoder-based models, typically trained via MLM, struggle to learn more complex/deeper CFGs.

## 5 RESULTS 6-9: HOW DO TRANSFORMERS LEARN NTS?

We now delve into the attention patterns. We demonstrate that these patterns mirror the CFG's syntactic structure and rules, with the transformer employing different attention heads to learn NTs at different CFG levels.

### 5.1 RESULT 6: POSITION-BASED ATTENTION

We first note that the transformer's attention weights are primarily influenced by the tokens' relative distance. This holds true even when *trained on the CFG data* with *absolute positional embedding*. This implies that the transformer learns the CFG's regularity and periodicity through positional information, which it then uses for generation. (We defer the figures to Appendix F.1 as this finding may not surprise some readers.) Motivated by this, we explore whether using position-based attention is *sufficient* to learn CFGs. In Figure 4, we find that $\text{GPT}_{\text{pos}}$ (or even $\text{GPT}_{\text{uni}}$) performs well, surpassing the vanilla `GPT`, but not reaching the full potential of $\text{GPT}_{\text{rel}}$. This supports the superior practical performance of relative-position based transformer variants (such as $\text{GPT}_{\text{rel}}$, $\text{GPT}_{\text{rot}}$, `deBERTa`) over their base models (`GPT` or `BERT`). On this other hand, this also indicates that **position-based attention alone is not enough for transformers to learn CFGs.**

## 5.2 RESULT 7-9: BOUNDARY-BASED ATTENTION

Next, we *remove* the position-bias from the attention matrix to examine the remaining part. We find that the transformer also learns a strong boundary-based attention pattern, where tokens on the NT-end boundaries typically **attend to the "most adjacent" NT-end boundaries**, see Figure 2. This attention pattern enables the transformer to effectively learn the hierarchical and recursive structure of the CFG, and generate output tokens based on the NT symbols and rules.

Formally, let $A_{l,h,j \to i}(x)$ for $j \geq i$ denote the attention weight for positions $j \to i$ at layer $l$ and head $h$ of the transformer, on input sequence $x$. Given a sample pool $\{x^{(n)}\}_{n \in [N]} \in L(\mathcal{G})$, we compute for each layer $l$, head $h$,[11]

$$\overline{A}_{l,h,p} = Average[\![A_{l,h,j \to i}(x^{(n)}) \mid n \in N, 1 \leq i \leq j \leq \mathbf{len}(x^{(n)}) \text{ s.t. } j - i = p]\!] \ ,$$

which represents the average attention between any token pairs of distance $p$ over the sample pool. To remove position-bias, we focus on $B_{l,h,j \to i}(x) := A_{l,h,j \to i}(x) - \overline{A}_{l,h,j-i}$ in this subsection. Our observation can be broken down into three steps.

**Result 7** (Figure 7(a)). $B_{l,h,j \to i}(x)$ *exhibits a strong bias towards* **tokens** $i$ **at NT ends**.

This can be seen in Figure 7(a), where we present the average value of $B_{l,h,j \to i}(x)$ over data $x$ and pairs $i, j$ where $i + \delta$ is the deepest NT-end at level $\ell$ (symbolically, $\mathfrak{b}^\sharp(i + \delta) = \ell$). The attention weights are highest when $\delta = 0$ and decrease rapidly for surrounding tokens.

**Result 8** (Figure 7(b)). $B_{l,h,j \to i}(x)$ *favors pairs* $i, j$ **both at NT ends** *at some level* $\ell$.

This can be seen in Figure 7(b), where we show the average $B_{l,h,j \to i}(x)$ over data $x$ and pairs $i, j$ where $\mathfrak{b}_\ell(i + \delta_1) = \mathfrak{b}_\ell(j + \delta_2) = 1$ for $\delta_1, \delta_2 \in \{-1, 0, 1\}$. It is maximized when $\delta_1 = \delta_2 = 0$.

**Result 9** (Figure 7(c)). $B_{l,h,j \to i}(x)$ *favors* **"adjacent" NT-end token pairs** $i, j$.

Above, we define "adjacency" as follows. We introduce $B^{\text{end} \to \text{end}}_{l,h,\ell' \to \ell,r}$ to represent the average value of $B_{l,h,j \to i}(x)$ over samples $x$ and token pairs $i, j$ that are at the deepest NT-ends on levels $\ell, \ell'$ respectively (symbolically, $\mathfrak{b}^\sharp(i) = \ell \wedge \mathfrak{b}^\sharp(j) = \ell'$), and are at a distance $r$ based on the ancestor indices at level $\ell$ (symbolically, $\mathfrak{p}_\ell(j) - \mathfrak{p}_\ell(i) = r$). We observe that $B^{\text{end} \to \text{end}}_{l,h,\ell' \to \ell,r}$ decreases as $r$ increases, and is highest when $r = 0$ (or $r = 1$ for pairs $\ell' \to \ell$ without an $r = 0$ entry).[12]

In conclusion, tokens corresponding to NT-ends at level $\ell'$ statistically have higher attention weights to their *most adjacent* NT-ends at every level $\ell$, *even after removing position-bias*.[13]

## 5.3 CONNECTION TO DP

Dynamic programming (DP) comprises two components: *storage* and *recurrent formula*. Identifying a specific DP implementation that a transformer follows is challenging due to the "exponentially many" ways to implement such DPs (see Footnote 9). However, we highlight *common elements* in all DP implementations and their correlation with the transformer. In Section 4, we demonstrated that transformers can encode the DP's *storage* "backbone", encompassing all necessary $\text{DP}(i, j, a)$ on the correct CFG parsing tree, regardless of the DP implementation.

For the *recurrent formula*, consider $\text{DP}(k, j, a)$ in the backbone, derived from $\text{DP}(k, i, b) \wedge \text{DP}(i, j, c)$ using CFG rule $a \mapsto b, c$. Given that $\text{DP}(k, i, b)$ is stored near position $i$ while $\text{DP}(k, j, a)$ and $\text{DP}(i, j, c)$ are stored near position $j$ (Result 5), the model needs to perform a *memory read* of position $i$ from position $j$, or $j \to i$. Note that positions $i$ and $j$ are adjacent NT-ends *of the same level*, and we have verified that GPT models favor attending $j \to i$ when $i$ and $j$ are adjacent NT-ends, serving as evidence that (decoder-only) transformers use a DP-like approach. See Figure 8 (top) for an illustration.

---

[11]Throughout this paper, we use $[\![\cdot]\!]$ to denote multi-sets that allow multiplicity, such as $[\![1, 2, 2, 3]\!]$. This allows us to conveniently talk about its set average.

[12]For any token pair $j \to i$ with $\ell = \mathfrak{b}^\sharp(i) \geq \mathfrak{b}^\sharp(j) = \ell'$ — meaning $i$ is at an NT-end closer to the root than $j$ — it satisfies $\mathfrak{p}_\ell(j) - \mathfrak{p}_\ell(i) \geq 1$ so their distance $r$ is strictly positive.

[13]Without removing position-bias, such a statement might be meaningless as the position-bias may favor "adjacent" anything, including NT-end pairs.

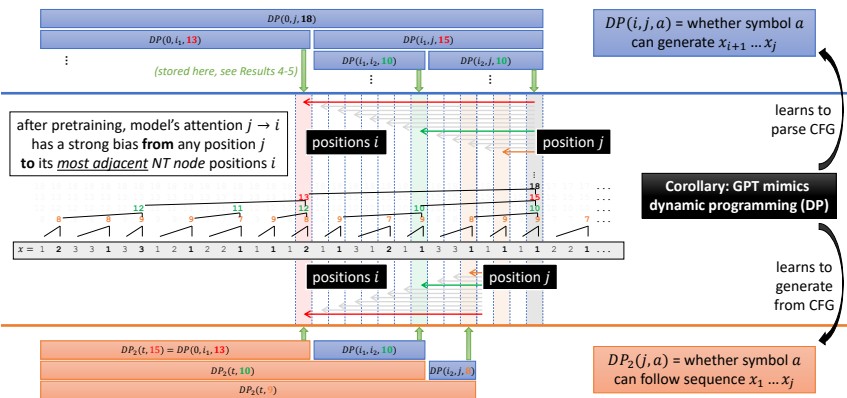

Figure 8: Illustration of how GPTs mimic dynamic programming. See discussions in Section 5.3.

**Further reading for experts.** Transformers are not only parsing algorithms but also generative ones. Experts in CFGs (or experienced participants in coding competitions) may immediately understand that the generative process requires implementing a second DP:

let $\text{DP}_2(j, a)$ denote if prefix $x_1, \ldots, x_j$ can be followed with a given symbol $a \in \mathbf{NT} \cup \mathbf{T}$.

Suppose there is a rule $b \mapsto c, a$, and $\text{DP}(i, j, c) \wedge \text{DP}_2(i, b)$ both hold; this implies $\text{DP}_2(j, a)$ also holds. This is analogous to the inside-outside algorithm (Baker, 1979), and is a special case of problem 6 in the IOI 2006 competition. In this case, the model also needs to perform a *memory read* of position $i$ from position $j$. Here, position $i$ is the most adjacent NT-end to position $j$ *at a different level*; we have *also* verified that GPT models favor attending such $j \to i$. See Figure 8 (bottom).

Finally, the above demonstration shows how to correctly parse and generate, but to generate following the same distribution of CFGs, the model needs to learn $\text{DP}_2'(j, a)$, the probability that symbol $a$ can follow prefix $x_1, \ldots, x_j$. The recurrent formula is similar in terms of memory read patterns (thus the attention patterns). We ignore this subtlety for conciseness.

In sum, while identifying a specific DP implementation that a transformer learns is nearly impossible, we have shown that the backbone of the DP — including the necessary DP storage states and recurrent formula — are observable in the pretrained models' hidden states and attention patterns. This serves as strong evidence that pretrained (decoder-only) transformers largely mimic dynamic programming, regardless of the specific DP implementation they choose.

## 6 RELATED WORK AND CONCLUSION

We defer *implicit CFGs* and *robust CFGs* to Appendix B.

Transformers can encode some CFGs, especially those that correspond to natural languages (Hewitt & Manning, 2019; Shi et al., 2022; Zhao et al., 2023; Maudslay & Cotterell, 2021; Manning et al., 2020; Vilares et al., 2020; Wu et al., 2020; Arps et al., 2022). Deletang et al. (2023) studied transformer's learnability on a few languages in the Chomsky hierarchy (which includes CFGs) However, the *inner mechanisms* regarding how transformer can or cannot solve those tasks are unclear. There are works "better" than us by precisely interpreting each neuron's function, but they study simpler tasks using simpler architectures. For instance, Nanda et al. (2023) examined 1 or 2-layer transformers with context length 3 for the arithmetic addition. In addition to linear probing, Murty et al. (2023) explored alternative methods to deduce the tree structures learned by a transformer. They developed a score to quantify the "tree-like" nature of a transformer, demonstrating that it becomes increasingly tree-like during training. Our Figure 20 in Appendix E.3 also confirmed on such findings. (This paper appears in May 2023, so we focus on related works before that.)

**Conclusion.** We studied how transformers learn synthetically generated, yet challenging CFGs, and show the inner workings correlate with the internal states of the dynamic programming algorithms needed to parse such CFGs. We hope this will point towards more opportunities towards understanding larger models on complex tasks. (Indeed, we are writing a series of papers using the findings and probing techniques developed from this paper; we cannot cite them due to anonymity.)

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

# APPENDIX

## A  MISSING FIGURE

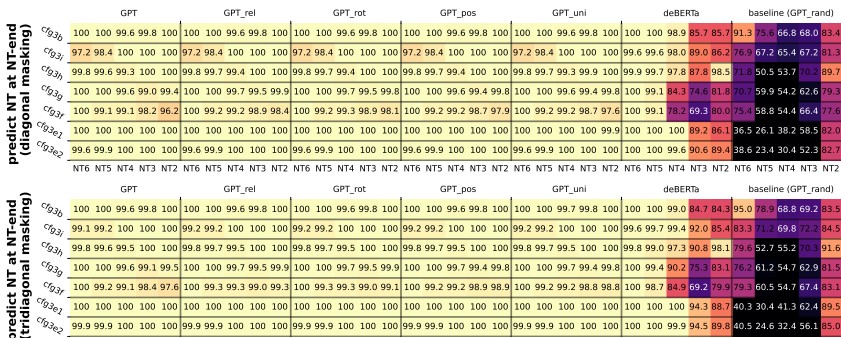

Figure 9: Generative models encode NT ancestors **almost exactly at** NT boundaries.  The $NT_\ell$ column represents the accuracy to predict $\mathfrak{s}_\ell(i)$ at locations $i$ with $\mathfrak{b}_\ell(i) = 1$, via diagonal multi-head linear probing (4.3).

---

**Observation.**  By comparing against a baseline, which is the encoding from a random GPT, we see that BERT-like (encoder-only) transformers such as DeBERTa trained on a masked language modeling (MLM) task, do not store deep NT ancestor information at the NT boundaries.

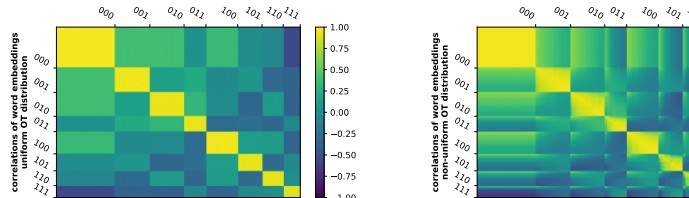

Figure 10: Language models learn implicit CFGs by using word embeddings to encode the (hidden) terminal symbol.

We present word embedding correlations for GPT pre-trained on an implicit CFG with $|\mathbf{T}| = 3$ and vocabulary size $|\mathbf{OT}| = 300$. There are 300 rows/columns each representing an observable token $a \in \mathbf{OT}$. Label $ijk \in \{0,1\}^3$ in the figure indicates whether $a$ is in $\mathbf{OT}_t$ for the three choices $t \in \mathbf{T}$. Details are in Section B.1.

## B   RESULTS 10-13: EXTENSIONS OF CFGS

### B.1   RESULT 10: IMPLICIT CFGS

In an *implicit CFG*, terminal symbols represent bags of tokens with shared properties. For example, a terminal symbol like $noun$ corresponds to a distribution over a bag of nouns, while $verb$ corresponds to a distribution over a bag of verbs. These distributions can be non-uniform and overlapping, allowing tokens to be shared between different terminal symbols. During pre-training, the model learns to associate tokens with their respective syntactic or semantic categories, without prior knowledge of their specific roles in the CFG.

Formally, we consider a set of *observable tokens* $\mathbf{OT}$, and each terminal symbol $t \in \mathbf{T}$ in $\mathcal{G}$ is associated with a subset $\mathbf{OT}_t \subseteq \mathbf{OT}$ and a probability distribution $\mathcal{D}_t$ over $\mathbf{OT}_t$. The sets $(\mathbf{OT}_t)_t$ can be overlapping. To generate a string from this implicit CFG, after generating $x = (x_1, x_2, \ldots, x_m) \sim L(\mathcal{G})$, for each terminal symbol $x_i$, we independently sample one element $y_i \sim \mathcal{D}_{x_i}$. After that, we observe the new string $y = (y_1, y_2, \cdots, y_m)$, and let this new distribution be called $y \sim L_O(\mathcal{G})$

We pre-train language models using samples from the distribution $y \sim L_O(\mathcal{G})$. During testing, we evaluate the success probability of the model generating a string that belongs to $L_O(\mathcal{G})$, given an input prefix $y_{:c}$. Or, in symbols,

$$\mathbf{Pr}_{y \sim L_O(\mathcal{G}) + \text{randomness of } F} \left[ (y_{:c}, F(y_{:c})) \in L_O(\mathcal{G}) \right] \ ,$$

where $F(y_{:c})$ represents the model's generated completion given prefix $y_{:c}$. (We again use dynamic programming to determine whether the output string is in $L_O(\mathcal{G})$.)

We summarize our finding below and deferring details to Appendix G.

**Result 10** (Figure 10). *Generative language models can learn implicit CFGs very well. In particular, after pretraining, the token embeddings from the same subset $\mathbf{OT}_t$ are grouped together, indicating they use token embedding layer to encode the hidden terminal symbol information.*

### B.2   RESULTS 11-13: ROBUSTNESS ON CORRUPTED CFG

One may also wish to pre-train a transformer to be *robust* against errors and inconsistencies in the input. For example, if the input data is a prefix with some tokens being corrupted or missing, then one may hope the transformer to correct the errors and still complete the sentence following the correct CFG rules. Robustness is an important property, as it reflects the generalization and adaptation ability of the transformer to deal with real-world training data, which may not always follow the CFG perfectly (such as having grammar errors).

To test robustness, for each input prefix $x_{:c}$ of length $c$ that belongs to the CFG, we randomly select a set of positions $i \in [c]$ in this prefix — each with probability $\rho$ — and flip them i.i.d. with a random symbol in $\mathbf{T}$. Call the resulting prefix $\widetilde{x}_{:c}$. Next, we feed the *corrupted prefix* $\widetilde{x}_{:c}$ to the transformer $F$ and compute its generation accuracy in the uncorrupted CFG: $\mathbf{Pr}_{x \sim L(\mathcal{G}), F}[(x_{:c}, F(\widetilde{x}_{:c})) \in L(\mathcal{G})]$.

| | pre-training method | | |
| --- | --- | --- | --- |
| generation acc (%) for cfg3b | NT-level 0.1 random perturbation | T-level 0.15 random perturbation | NT-level 0.05 deterministic permutation |
| cut0 τ=0.1 | 100 100 100 100 100 100 100 100 100 100 | 100 100 100 100 100 100 100 100 100 | 99.8 100 100 100 100 100 100 100 100 100 · 100 |
| cut0 τ=0.2 | 98.7 100 100 100 100 100 100 100 100 100 | 99.2 99.9 100 100 100 99.9 100 100 100 100 | 98.5 100 100 100 100 100 100 100 100 100 · 100 |
| cut0 τ=1 | 0.0 14.3 24.7 39.8 44.4 55.7 64.5 73.5 82.6 91.8 | 0.0 14.1 22.8 35.3 44.9 58.2 65.4 75.5 83.6 92.5 | 0.0 14.7 26.9 38.5 49.8 56.8 65.5 75.2 81.5 91.8 · 99.8 |
| corrupted cut50 τ=0.1 | 78.3 78.9 80.6 78.0 79.1 78.6 79.5 78.6 76.4 77.9 | 82.6 80.4 80.6 80.4 81.7 82.6 81.4 81.7 80.8 80.8 | 60.4 58.3 56.5 58.1 60.4 59.1 60.6 57.5 58.9 56.9 · 30.0 |
| corrupted cut50 τ=0.2 | 77.4 78.7 80.0 76.6 77.8 78.2 78.3 77.3 74.9 77.9 | 81.1 81.1 80.5 79.6 81.2 82.0 81.4 80.7 80.0 80.4 | 59.5 57.7 55.9 57.6 59.2 58.8 59.7 57.2 57.8 57.1 · 30.3 |
| corrupted cut50 τ=1 | 0.0 0.5 0.5 0.6 0.5 0.3 0.6 0.4 0.5 0.7 | 0.0 0.4 0.5 0.8 0.2 0.3 0.5 0.6 0.7 0.6 | 0.0 0.1 0.4 0.4 0.4 0.5 0.9 0.5 0.3 0.3 · 29.6 |
| cut50 τ=0.1 | 100 100 100 100 100 100 100 100 100 100 | 100 100 100 100 100 100 100 100 100 100 | 99.4 100 100 100 100 100 100 100 100 100 · 100 |
| cut50 τ=0.2 | 99.2 100 100 100 100 100 100 100 100 100 | 99.6 100 100 100 100 100 100 100 100 100 | 98.4 100 100 100 100 100 100 100 100 100 · 100 |
| cut50 τ=1 | 0.0 91.5 95.7 97.1 98.1 98.7 99.2 99.0 99.5 99.4 | 0.0 92.8 96.2 97.6 98.2 99.1 99.3 99.4 99.5 99.7 | 0.0 83.4 90.6 94.0 96.2 97.2 98.1 98.7 99.2 99.3 · 99.9 |
| | 1.0 0.9 0.8 0.7 0.6 0.5 0.4 0.3 0.2 0.1 | 1.0 0.9 0.8 0.7 0.6 0.5 0.4 0.3 0.2 0.1 | 1.0 0.9 0.8 0.7 0.6 0.5 0.4 0.3 0.2 0.1 clean |

pre-training data perturbation ratio $\gamma$ OR clean data

Figure 11: Generation accuracies for models pre-trained cleanly VS pre-trained over perturbed data, on clean or corrupted prefixes with cuts $c = 0$ or $c = 50$, using generation temperatures $\tau = 0.1, 0.2, 1.0$.

---

**Observation.** In Rows 4/5, by comparing against the last column, we see it is *beneficial* to include low-quality data (e.g. grammar mistakes) during pre-training. The amount of low-quality data could be little ($\gamma = 0.1$ fraction) or large (*every training sentence may have grammar mistake*). The transformer also learns a "mode switch" between the "correct mode" or not; details in Section B.2.

---

We not only consider clean pre-training, but also some versions of *robust pre-training*. That is, we randomly select $\gamma \in [0, 1]$ fraction of the training data and perturb them before feeding into the pre-training process. We compare three types of data perturbations.[14]

- (T-level random perturbation). Each $x_i$ w.p. 0.15 we replace it with a random symbol in $\mathbf{T}$.
- (NT-level random perturbation). Let $\ell = L - 1$ and recall $s_\ell = \left(s_{\ell,1}, s_{\ell,2}, \ldots, s_{\ell,m_{L-1}}\right)$ is the sequence of symbols at NT-level $\ell$. For each $s_{\ell,i}$, w.p. 0.10 we perturb it to a random symbol in $\mathbf{NT}_\ell$; and then generate $x = s_L$ according to this perturbed sequence.
- (NT-level deterministic perturbation). Let $\ell = L - 1$ and fix a permutation $\pi$ over symbols in $\mathbf{NT}_\ell$. For each $s_{\ell,i}$, w.p. 0.05 we perturb it to its next symbol in $\mathbf{NT}_{L-1}$ according to $\pi$; and then generate $x = s_L$ according to this perturbed sequence.

We focus on $\rho = 0.15$ with a wide range of perturbation rate $\tau = 0.0, 0.1, \ldots, 0.9, 1.0$. We present our findings in Figure 11. The main message is:

**Result 11** (Figure 11, rows 4/5). *When pretrained over clean data, GPT models are **not so robust** to "grammar mistakes." It is **beneficial** to include corrupted or low-quality pretrain data.*

Specifically, GPT models achieve only $\sim 30\%$ accuracy when pretrained over clean data $x \sim L(\mathcal{G})$. If we pretrain from perturbed data — *both* when $\gamma = 1.0$ so all data are perturbed, *and* when $\gamma = 0.1$ so we have a small fraction of perturbed data — GPT can achieve $\sim 79\%, 82\%$ and $60\%$ robust accuracies respectively using the three types of data perturbations (rows 4/5 of Figure 11).

Next, we take a closer look. If we use temperature $\tau = 1$ for generation:

**Result 12** (Figure 11, rows 3/6/9). *Pre-training on corrupted data teaches model a **mode switch**.*

- *Given a correct prefix, it mostly completes with a correct string in the CFG (Row 9);*

- *Given a corrupted prefix, it **always** completes sentences with grammar mistakes (Row 6);*

- *When given no prefix, it generates corrupted strings with probability close to $\gamma$ (Row 3).*

By comparing the generation accuracies across different $\tau$ and $\gamma$, we observe:

**Result 13** (Figure 11, rows 4/5/6). *High robust accuracy is achieved when generating using low temperatures $\tau$,[15] and is not sensitive to $\gamma$ – the fraction of pretrain data that is perturbed.*

This should not be surprising given that the language model learned a "mode switch." Using low temperature encourages the model to, for each next token, pick a more probable solution. This allows it to achieve good robust accuracy *even when* the model is trained totally on corrupted data ($\gamma = 1.0$). Note this is consistent with practice: when feeding a pre-trained completion model (such

---

[14]One can easily extend our experiments by considering other types of data corruption (for evaluation), and other types of data perturbations (for training). We refrain from doing so because it is beyond the scope of this paper.

[15]Recall, when temperature $\tau = 0$ the generation is greedy and deterministic; when $\tau = 1$ it reflects the unaltered distribution learned by the transformer; when $\tau > 0$ s small it encourages the transformer to output "more probable" tokens.

as Llama or GPT-3-davinci003) with prompts of grammar mistakes, it tends to produce texts also with (even new!) grammar mistakes when using a large temperature.

Our experiments suggest that, additional instruct fine-tuning may be necessary, if one wants the model to *always* stay in the "correct mode" even for high temperatures. This is beyond the scope of this paper.

## C    EXPERIMENT SETUPS

### C.1    DATASET DETAILS

We construct seven synthetic CFGs of depth $L = 7$ with varying levels of learning difficulty. It can be inferred that the greater the number of T/NT symbols, the more challenging it is to learn the CFG. For this reason, to push the capabilities of language models to their limits, we primarily focus on cfg3b, cfg3i, cfg3h, cfg3g, cfg3f, which are of sizes $(1, 3, 3, 3, 3, 3, 3)$ and present increasing levels of difficulty. Detailed information about these CFGs is provided in Figure 12:

- In cfg3b, we construct the CFG such that the degree $|\mathcal{R}(a)| = 2$ for every NT $a$. We also ensure that in any generation rule, consecutive pairs of T/NT symbols are distinct.

  The 25%, 50%, 75%, and 95% percentile string lengths are $251, 278, 308, 342$ respectively.

- In cfg3i, we set $|\mathcal{R}(a)| = 2$ for every NT $a$. We remove the requirement for distinctness to make the data more challenging than cfg3b.

  The 25%, 50%, 75%, and 95% percentile string lengths are $276, 307, 340, 386$ respectively.

- In cfg3h, we set $|\mathcal{R}(a)| \in \{2, 3\}$ for every NT $a$ to make the data more challenging than cfg3i.

  The 25%, 50%, 75%, and 95% percentile string lengths are $202, 238, 270, 300$ respectively.

- In cfg3g, we set $|\mathcal{R}(a)| = 3$ for every NT $a$ to make the data more challenging than cfg3h.

  The 25%, 50%, 75%, and 95% percentile string lengths are $212, 258, 294, 341$ respectively.

- In cfg3f, we set $|\mathcal{R}(a)| \in \{3, 4\}$ for every NT $a$ to make the data more challenging than cfg3g.

  The 25%, 50%, 75%, and 95% percentile string lengths are $191, 247, 302, 364$ respectively.

*Remark* C.1.  From the examples in Figure 12, it becomes evident that for grammars $\mathcal{G}$ of depth 7, proving that a string $x$ belongs to $L(\mathcal{G})$ is highly non-trivial, even for a human being, and even when the CFG rules are known. The standard method of demonstrating $x \in L(\mathcal{G})$ is through dynamic programming. We further discuss what we mean by a CFG's "difficulty" in Appendix I, and provide additional experiments beyond the cfg3 data family.

*Remark* C.2.  cfg3f is a dataset that sits right on the boundary of difficulty at which GPT2-small is capable of learning, see Figure 30 later which shows that smaller GPT2 cannot learn such cfg3f (and refer to subsequent subsections for training parameters). While it is certainly possible to consider deeper and more complex CFGs, this would necessitate training a larger network for a longer period. We choose not to do this as our findings are sufficiently convincing at the level of cfg3f.

Simultaneously, to illustrate that transformers can learn CFGs with larger $|\mathbf{NT}|$ or $|\mathbf{T}|$, we construct datasets cfg3e1 and cfg3e2 respectively of sizes $(1, 3, 9, 27, 81, 27, 9)$ and $(1, 3, 9, 27, 27, 9, 4)$. They are too lengthy to describe so only included in the supplementary materials.

### C.2    MODEL ARCHITECTURE DETAILS

We define `GPT` as the standard GPT2-small architecture (Radford et al., 2019), which consists of 12 layers, 12 attention heads per layer, and 768 ($=12 \times 64$) hidden dimensions. We pre-train `GPT` on the aforementioned datasets, starting from random initialization. For a baseline comparison, we also implement DeBERTa (He et al., 2020), resizing it to match the dimensions of GPT2 — thus also comprising 12 layers, 12 attention heads, and 768 dimensions.

**Architecture size.**    We have experimented with models of varying sizes and observed that their learning capabilities scale with the complexity of the CFGs. To ensure a fair comparison and enhance reproducibility, we primarily focus on models with 12 layers, 12 attention heads, and 768 dimensions. The transformers constructed in this manner consist of 86M parameters.

**Modern GPTs with relative attention.**    Recent research (He et al., 2020; Su et al., 2021; Black et al., 2022) has demonstrated that transformers can significantly improve performance by using

Figure 12: The context-free grammars cfg3b, cfg3i, cfg3h, cfg3g, cfg3f that we primarily use in this paper, together with a sample string from each of them.

**Observation.** Although those CFGs are only of depth 7, they are capable of generating sufficiently long and hard instances; after all, even when the CFG rules are given, the typical way to decide if a string $x$ belongs to the CFG language $x \in L(\mathcal{G})$ may require dynamic programming.

attention mechanisms based on the *relative* position differences of tokens, as opposed to the absolute positions used in the original GPT2 (Radford et al., 2019) or BERT (Kenton & Toutanova, 2019). There are two main approaches to achieve this. The first is to use a "relative positional embedding layer" on $|j - i|$ when calculating the attention from $j$ to $i$ (or a bucket embedding to save space). This approach is the most effective but tends to train slower. The second approach is to apply a rotary positional embedding (RoPE) transformation (Su et al., 2021) on the hidden states; this is known to be slightly less effective than the relative approach, but it can be trained much faster.

We have implemented both approaches. We adopted the RoPE implementation from the GPT-NeoX-20B project (along with the default parameters), but downsized it to fit the GPT2 small model. We refer to this architecture as $\mathtt{GPT_{rot}}$. Since we could not find a standard implementation of GPT using relative attention, we re-implemented GPT2 using the relative attention framework from DeBERTa (He et al., 2020). (Recall, DeBERTa is a variant of BERT that effectively utilizes relative positional embeddings.) We refer to this architecture as $\mathtt{GPT_{rel}}$.

**Weaker GPTs utilizing only position-based attention.** For the purpose of analysis, we also consider two significantly weaker variants of GPT, where the attention matrix *exclusively depends* on the token positions, and not on the input sequences or hidden embeddings. In other words, the attention pattern remains *constant* for all input sequences.

We implement $\mathtt{GPT_{pos}}$, a variant of $\mathtt{GPT_{rel}}$ that restricts the attention matrix to be computed solely using the (trainable) relative positional embedding. This can be perceived as a GPT variant that *maximizes the use of position-based attention*. We also implement $\mathtt{GPT_{uni}}$, a 12-layer, 8-head, 1024-dimension transformer, where the attention matrix is *fixed*; for each $h \in [8]$, the $h$-th head *consistently* uses a fixed, uniform attention over the previous $2^h - 1$ tokens. This can be perceived as a GPT variant that *employs the simplest form of position-based attention*.

*Remark* C.3. It should not be surprising that $\mathtt{GPT_{pos}}$ or $\mathtt{GPT_{uni}}$ perform much worse than other GPT models on real-life wikibook pre-training. However, once again, we use them only for *analysis*

*purpose* in this paper, as we wish to demonstrate what is the maximum power of GPT when only using position-based attention to learn CFGs, and what is the marginal effect when one goes *beyond* position-based attention.

**Features from random transformer.** Finally we also consider a randomly-initialized $\mathrm{GPT_{rel}}$, and use those random features for the purpose of predicting NT ancestors and NT ends. This serves as a baseline, and can be viewed as the power of the so-called (finite-width) neural tangent kernel (Jacot et al., 2018; Allen-Zhu et al., 2019). We call this $\mathrm{GPT_{rand}}$.

### C.3 PRE-TRAINING DETAILS

For each sample $x \sim L(\mathcal{G})$ we append it to the left with a BOS token and to the right with an EOS token. Then, following the tradition of language modeling (LM) pre-training, we concatenate consecutive samples and randomly cut the data to form sequences of a fixed window length 512.

As a baseline comparison, we also applied DeBERTa on a masked language modeling (MLM) task for our datasets. We use standard MLM parameters: 15% masked probability, in which 80% chance of using a masked token, 10% chance using the original token, and 10% chance using a random token.

We use standard initializations from the huggingface library. For GPT pre-training, we use AdamW with $\beta = (0.9, 0.98)$, weight decay 0.1, learning rate 0.0003, and batch size 96. We pre-train the model for 100k iterations, with a linear learning rate decay.[16] For DeBERTa, we use learning rate 0.0001 which is better and 2000 steps of learning rate linear warmup.

Throughout the experiments, for both pre-training and testing, we only use **fresh samples** from the CFG datasets (thus using 4.9 billion tokens = $96 \times 512 \times 100k$). We have also tested pre-training with a finite training set of $100m$ tokens; and the conclusions of this paper stay similar. To make this paper clean, we choose to stick to the infinite-data regime in this version of the paper, because it enables us to make negative statements (for instance about the vanilla GPT or DeBERTa, or about the learnability of NT ancestors / NT boundaries) without worrying about the sample size. Please note, given that our CFG language is very large (e.g., length 300 tree of length-2/3 rules and degree 4 would have at least $4^{300/3}$ possibility), there is *almost no chance that training/testing hit the same sentence*.

As for the reproducibility of our result, we did not run each pre-train experiment more than once (or plot any confidence interval). This is because, rather than repeating our experiments identically, we find it more interesting to use the resources to run it against different datasets and against different parameters. We pick the best model using the perplexity score from each pre-training task. When evaluating the generation accuracy in Figure 4, we have generated more than 20000 samples for each case, and present the diversity pattern accordingly in Figure 13.

### C.4 PREDICT NT ANCESTOR AND NT BOUNDARY

Recall from Section 4.1 that we have proposed to use a multi-head linear function to probe whether or not the hidden states of a transformer, implicitly encodes the NT ancestor and NT boundary information for each token position. Since this linear function can be of dimension $512 \times 768$ — when having a context length 512 and hidden dimension 768 — recall in (4.2), we have proposed to use a multi-head attention to construct such linear function for efficient learning purpose. This significantly reduces sample complexity and makes it much easier to find the linear function.

In our implementation, we choose $H = 16$ heads and hidden dimension $d' = 1024$ when constructing this position-based attention in (4.2). We have also tried other parameters but the NT ancestor/boundary prediction accuracies are not very sensitive to such architecture change. We again use AdamW with $\beta = (0.9, 0.98)$ but this time with learning rate 0.003, weight decay 0.001, batch size 60 and train for 30k iterations.

Once again we use *fresh new samples* when training such linear functions. When evaluating the accuracies on predicting the NT ancester / boundary information, we also use fresh new samples. Recall our CFG language is sufficiently large so there is negligible chance that the model has seen

---

[16] We have slightly tuned the parameters to make pre-training go best. We noticed for training GPTs over our CFG data, a warmup learning rate schedule is not needed.

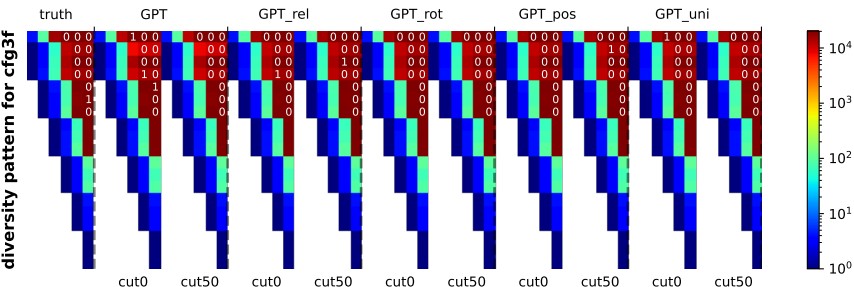

Figure 13: Comparing the generation diversity $\mathcal{S}^{\text{truth}}_{a\to\ell_2}$ and $\mathcal{S}^{F}_{a\to\ell_2}$ across different learned GPT models ($c=0$ or $c=50$). Rows correspond to NT symbols $a$ and columns correspond to $\ell_2 = 2, 3, \ldots, 7$. Colors represent the number of distinct elements in $\mathcal{S}^{\text{truth}}_{a\to\ell_2}$, and the white numbers represent the collision counts (if not present, meaning there are more than 5 collisions). More experiments in Figure 14, 15, and 16

---

**Observation.** We use $M = 20000$ samples. The diversity pattern from the pre-trained transformer matches that of the ground-truth. For instance, from the root one can generate $\Omega(M^2)$ distinct sequences to level $\ell_2 = 5$ using the CFG rules, and from every $a \in \mathbf{NT}_2$ one can generate $\Omega(M^2)$ to level $\ell_2 = 6$ (not to say to the T-level $\ell_2 = 7$); this is already more than the number of parameters in the model. Therefore, we conclude that the pre-trained model **does not rely on simply memorizing** a small set of patterns to learn the CFGs.

such a string during training.

# D MORE EXPERIMENTS ON GENERATION

Diversity can be estimated through entropy. Given a distribution $p$ over strings and a sampled subset $S = \{x^{(i)}\}_{i\in[M]}$ from $p$, for any string $x \in S$, denote by $\mathbf{len}(x)$ its length so $x = (x_1, \ldots, x_{\mathbf{len}(x)})$, and denote by $x_{\mathbf{len}(x)+1} = \mathsf{eos}$. The entropy in bits for $p$ can be estimated by

$$-\frac{1}{|S|} \sum_{x\in S} \sum_{i\in[\mathbf{len}(x)+1]} \log_2 \mathbf{Pr}_p\left[x_i \mid x_1, \ldots, x_{i-1}\right]$$

We compare the entropy of the true CFG distribution and the transformer's output distribution using $M = 20000$ samples in Figure 4 (middle).

Diversity can also be estimated using the birthday paradox to lower bound the support size of a distribution (Arora & Zhang, 2017). Given a distribution $p$ over strings and a sampled subset $S = \{x^{(i)}\}_{i\in[M]}$ from $p$, if every pair of samples in $S$ are distinct, then with good probability the support of $p$ is of size at least $\Omega(M^2)$. In Appendix D.1, we conducted an experiment with $M = 20000$. We performed a birthday paradox experiment from every symbol $a \in \mathbf{NT}_{\ell_1}$ to some other level $\ell_2 > \ell_1$, comparing that with the ground truth. For instance, we confirmed for the cfg3f dataset, there are at least $\Omega(M^2)$ distinct sentential forms that can be derived from a symbol in level 1 to level 5, or from level 2 to level 6, etc. — not to mention from the root in $\mathbf{NT}_1$ to the leaf at level 7. In particular, $M^2$ is already more than the number of parameters in the model.

From both experiments, we conclude that the pre-trained model **does not rely on simply memorizing** a small set of patterns to learn the CFGs.

## D.1 GENERATION DIVERSITY VIA BIRTHDAY PARADOX

Since "diversity" is influenced by the length of the input prefix, the length of the output, and the CFG rules, we want to carefully define what we measure.

Given a sample pool $x^{(1)}, \ldots, x^{(M)} \in L(\mathcal{G})$, for every symbol $a \in \mathbf{NT}_{\ell_1}$ and some later level $\ell_2 \geq \ell_1$ that is closer to the leaves, we wish to define a *multi-set* $\mathcal{S}_{a\to\ell_2}$ that describes *all possible generations from $a \in \mathbf{NT}_{\ell_1}$ to $\mathbf{NT}_{\ell_2}$* in this sample pool. Formally,

**Definition D.1.** *For $x \in L(\mathcal{G})$ and $\ell \in [L]$, we use $\mathfrak{s}_\ell(i..j)$ to denote the sequence of NT ancestor symbols at level $\ell \in [L]$ from position $i$ to $j$ with distinct ancestor indices:[17]*

$$\mathfrak{s}_\ell(i..j) = (\mathfrak{s}_\ell(k))_{k \in \{i,i+1,\ldots,j\} \text{ s.t. } \mathfrak{p}_\ell(k) \neq \mathfrak{p}_\ell(k+1)}$$

**Definition D.2.** *For symbol $a \in \mathbf{NT}_{\ell_1}$ and some layer $\ell_2 \in \{\ell_1, \ell_1 + 1, \ldots, L\}$, define multi-set[18]*

$$\mathcal{S}_{a \to \ell_2}(x) = \left[\!\!\left[\, \mathfrak{s}_{\ell_2}(i..j) \,\middle|\, \forall i, j, i \leq j \text{ such that } \mathfrak{p}_{\ell_1}(i-1) \neq \mathfrak{p}_{\ell_1}(i) = \mathfrak{p}_{\ell_1}(j) \neq \mathfrak{p}_{\ell_1}(j+1) \land a = \mathfrak{s}_{\ell_1}(i) \right]\!\!\right]$$

*and we define the multi-set union $\mathcal{S}_{a \to \ell_2} = \bigcup_{i \in [M]} \mathcal{S}_{a \to \ell_2}(x^{(i)})$, which is* **the multiset of all sentential forms that can be derived from NT symbol $a$ to depth $\ell_2$.**

(Above, when $x \sim L(\mathcal{G})$ is generated from the ground-truth CFG, then the ancestor indices and symbols $\mathfrak{p}, \mathfrak{s}$ are defined in Section 2. If $x \in L(\mathcal{G})$ is an output from the transformer $F$, then we let $\mathfrak{p}, \mathfrak{s}$ be computed using dynamic programming, breaking ties lexicographically.)

We use $\mathcal{S}_{a \to \ell_2}^{\text{truth}}$ to denote the ground truth $\mathcal{S}_{a \to \ell_2}$ when $x^{(1)}, \ldots, x^{(M)}$ are i.i.d. sampled from the real distribution $L(\mathcal{G})$, and denote by

$$\mathcal{S}_{a \to \ell_2}^{F} = \bigcup_{i \in [M'] \text{ and } x_{:c}^{(i)}, F(x_{:c}^{(i)}) \in L(\mathcal{G})} \mathcal{S}_{a \to \ell_2}\left(x_{:c}^{(i)}, F(x_{:c}^{(i)})\right)$$

that from the transformer $F$. For a fair comparison, for each $F$ and $p$, we pick an $M' \geq M$ such that $M = \left| \left\{ i \in [M'] \mid x_{:p}^{(i)}, F(x_{:p}^{(i)}) \in L(\mathcal{G}) \right\} \right|$ so that $F$ is capable of generating exactly $M$ sentences that nearly-perfectly satisfy the CFG rules.[19]

Intuitively, for $x$'s generated by the transformer model, the larger the number of distinct sequences in $\mathcal{S}_{a \to \ell_2}^{F}$ is, the more diverse the set of NTs at level $\ell_2$ (or Ts if $\ell_2 = L$) the model can generate starting from NT $a$. Moreover, in the event that $\mathcal{S}_{a \to \ell_2}^{F}$ has only distinct sequences (so collision count = 0), then we know that the generation from $a \to \ell_2$, with good probability, should include at least $\Omega(M^2)$ possibilities using a birthday paradox argument. [20]

For such reason, it can be beneficial if we compare the *number of distinct sequences* and the *collision counts* between $\mathcal{S}_{a \to \ell_2}^{F}$ and $\mathcal{S}_{a \to \ell_2}^{\text{truth}}$. Note we consider all $\ell_2 \geq \ell_1$ instead of only $\ell_2 = L$, because we want to better capture model's diversity at all CFG levels.[21] We present our findings in Figure 13 with $M = 20000$ samples for the cfg3f dataset.

In Figure 14 we present that for cfg3b, cfg3i, cfg3h, cfg3g, in Figure 15 for cfg3e1, and in Figure 16 for cfg3e2. We note that not only for hard, ambiguous datasets, also for those less ambiguous (cfg3e1, cfg3e2) datasets, language models are capable of generating very diverse outputs.

---

[17] With the understanding that $\mathfrak{p}_\ell(0) = \mathfrak{p}_\ell(\mathbf{len}(x) + 1) = \infty$.

[18] Throughout this paper, we use $[\![\cdot]\!]$ to denote multi-sets that allow multiplicity, such as $[\![1, 2, 2, 3]\!]$. This allows us to conveniently talk about its collision count, number of distinct elements, and set average.

[19] Please note $M$ and $M'$ are roughly the same, given

[20] A CFG of depth $L$, even with constant degree and constant size, can generate $2^{2^{\Omega(L)}}$ distinct sequences.

[21] A model might generate a same NT symbol sequence $s_{L-1}$, and then generate different Ts randomly from each NT. In this way, the model still generates strings $x$'s with large diversity, but $\mathcal{S}_{a \to L-1}^{F}(x)$ is small. If $\mathcal{S}_{a \to \ell_2}^{F}$ is large for every $\ell_2$ and $a$, then the generation from the model is *truely diverse at any level of the CFG*.

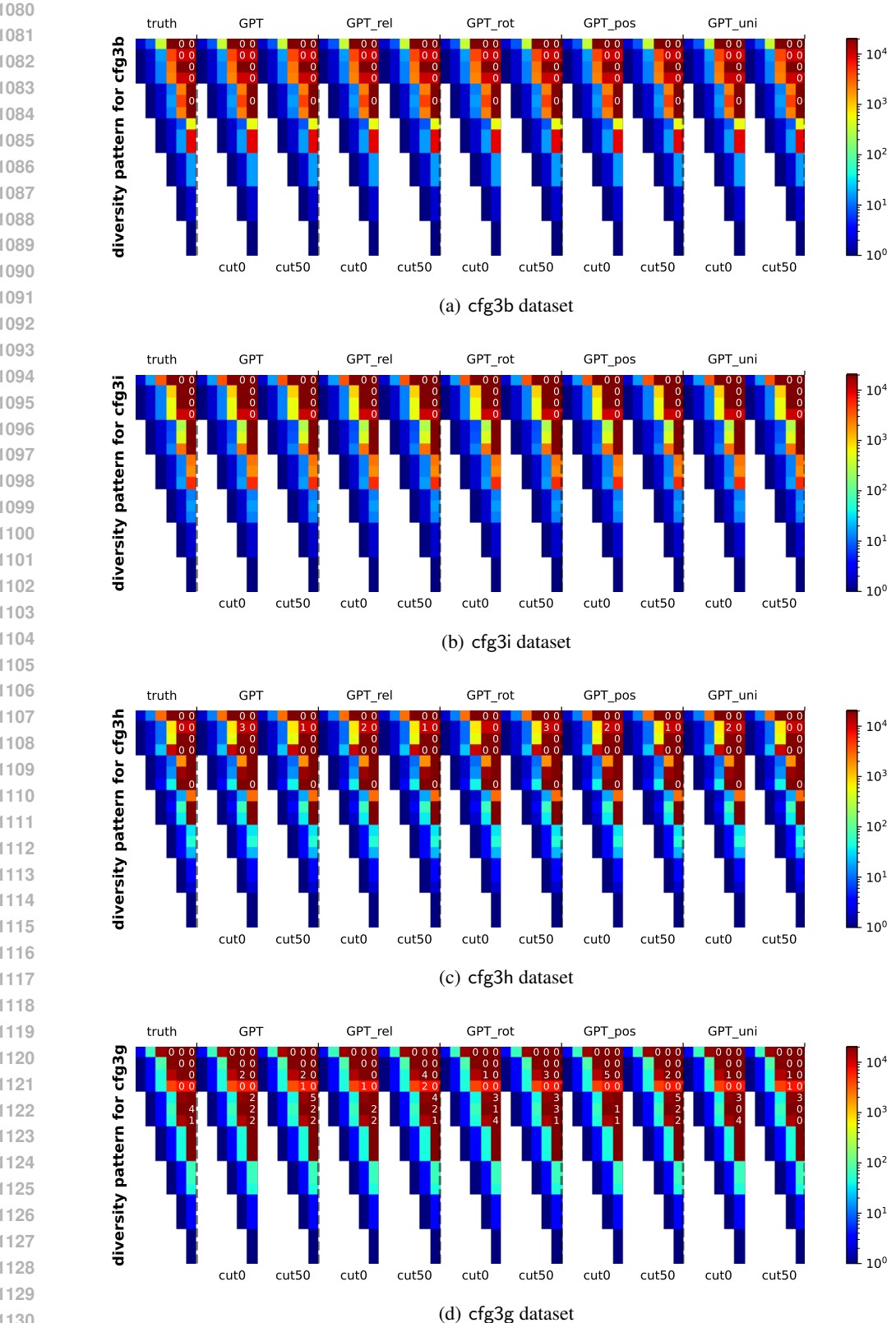

Figure 14: Comparing the generation diversity $\mathcal{S}_{a\to\ell_2}^{\text{truth}}$ and $\mathcal{S}_{a\to\ell_2}^{F}$ across different learned GPT models (and for $c = 0$ or $c = 50$). Rows correspond to NT symbols $a$ and columns correspond to $\ell_2 = 2, 3, \ldots, 7$. Colors represent the number of distinct elements in $\mathcal{S}_{a\to\ell_2}^{\text{truth}}$, and the white numbers represent the collision counts (if not present, meaning there are more than 5 collisions).

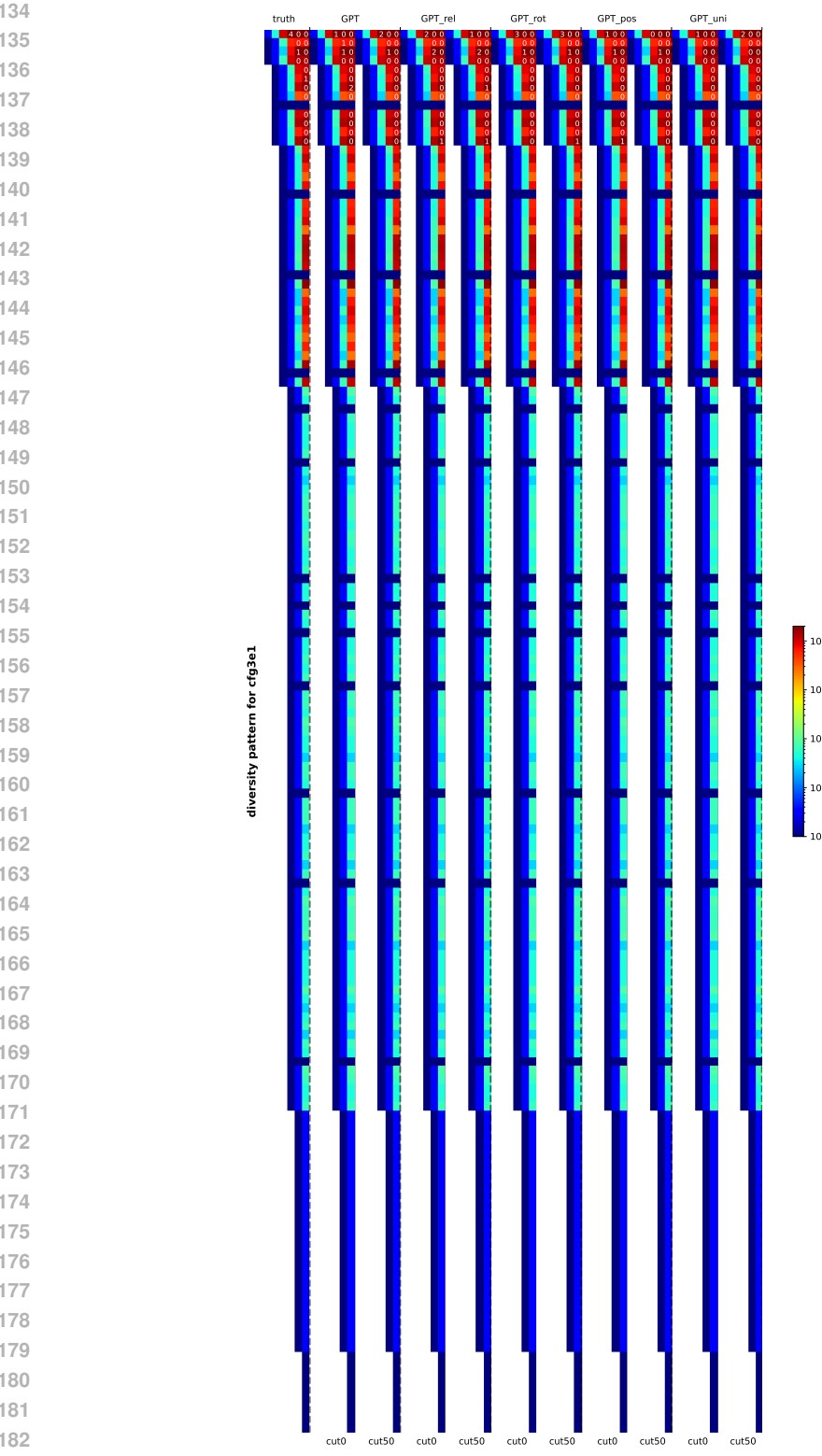

Figure 15: Comparing the generation diversity $\mathcal{S}_{a\to\ell_2}^{\text{truth}}$ and $\mathcal{S}_{a\to\ell_2}^{F}$ across different learned GPT models (and for $c = 0$ or $c = 50$). Rows correspond to NT symbols $a$ and columns correspond to $\ell_2 = 2, 3, \ldots, 7$. Colors represent the number of distinct elements in $\mathcal{S}_{a\to\ell_2}^{\text{truth}}$, and the white numbers represent the collision counts (if not present, meaning there are more than 5 collisions). This is for the cfg3e1 dataset.

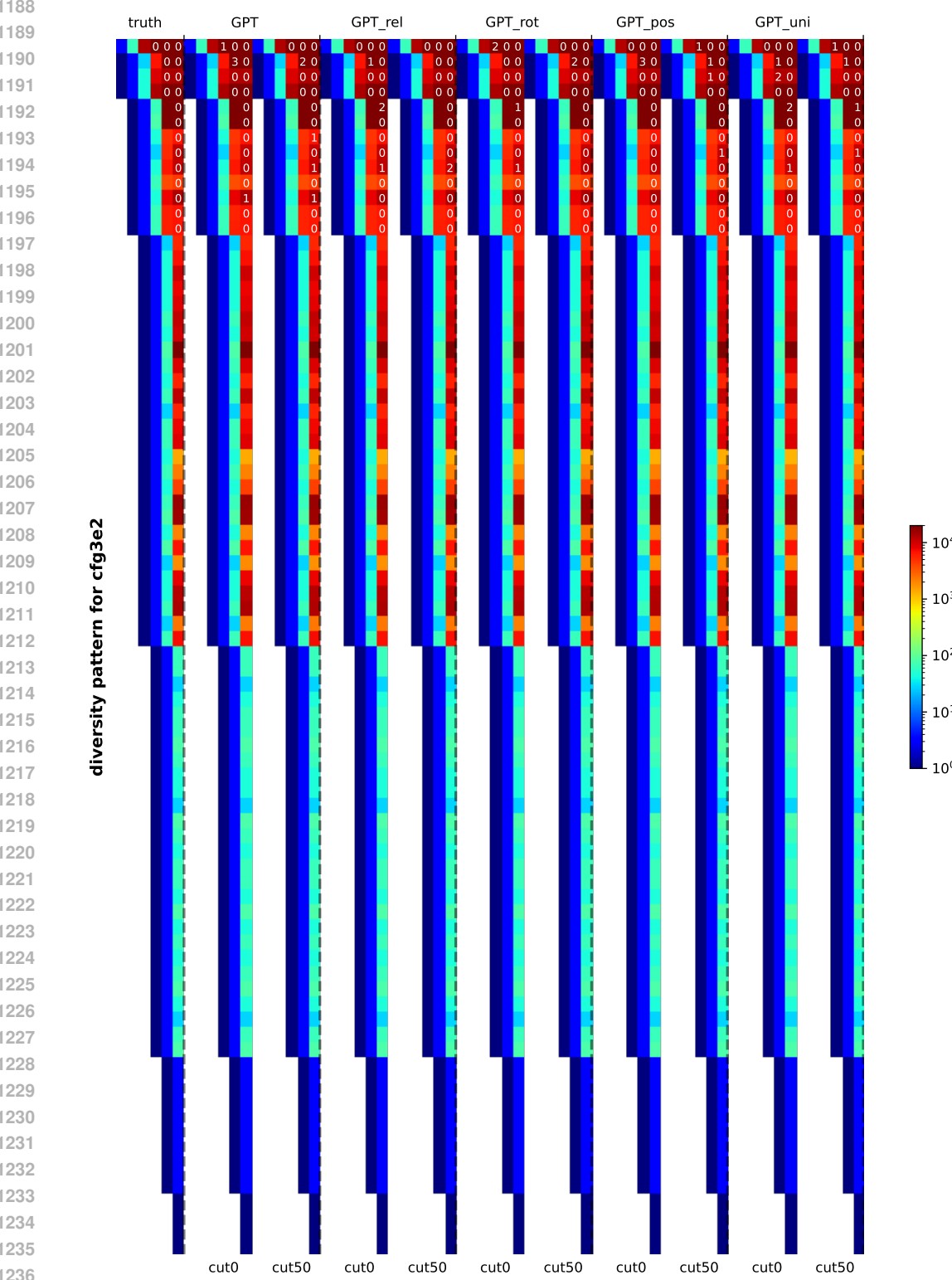

Figure 16: Comparing the generation diversity $\mathcal{S}^{\text{truth}}_{a\to\ell_2}$ and $\mathcal{S}^{F}_{a\to\ell_2}$ across different learned GPT models (and for $c = 0$ or $c = 50$). Rows correspond to NT symbols $a$ and columns correspond to $\ell_2 = 2, 3, \ldots, 7$. Colors represent the number of distinct elements in $\mathcal{S}^{\text{truth}}_{a\to\ell_2}$, and the white numbers represent the collision counts (if not present, meaning there are more than 5 collisions). This is for the cfg3e2 dataset.

## D.2 MARGINAL DISTRIBUTION COMPARISON

In order to effectively learn a CFG, it is also important to match the distribution of generating probabilities. While measuring this can be challenging, we have conducted at least a simple test on the marginal distributions $p(a, i)$, which represent the probability of symbol $a \in \mathbf{NT}_\ell$ appearing at position $i$ (i.e., the probability that $\mathfrak{s}_\ell(i) = a$). We observe a strong alignment between the generated probabilities and the ground-truth distribution. See Figure 17.

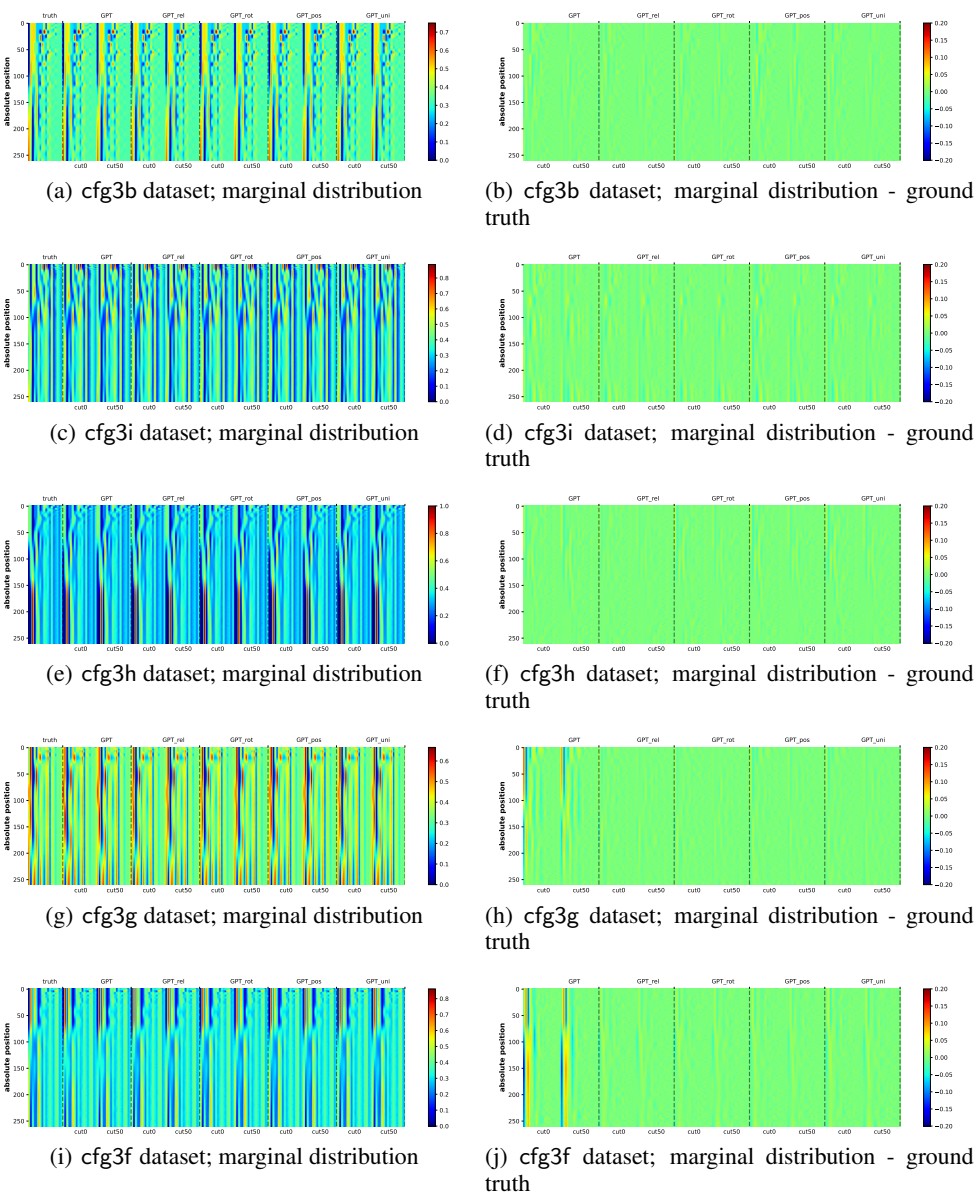

(a) cfg3b dataset; marginal distribution

(b) cfg3b dataset; marginal distribution - ground truth

(c) cfg3i dataset; marginal distribution

(d) cfg3i dataset; marginal distribution - ground truth

(e) cfg3h dataset; marginal distribution

(f) cfg3h dataset; marginal distribution - ground truth

(g) cfg3g dataset; marginal distribution

(h) cfg3g dataset; marginal distribution - ground truth

(i) cfg3f dataset; marginal distribution

(j) cfg3f dataset; marginal distribution - ground truth

Figure 17: Marginal distribution $p(a, i)$ difference between a trained model and the ground-truth, for an NT/T symbol $a$ (column) at position $i$ (row). Figures on the left compare the marginal distribution of the ground-truth against those generated from 5 models $\times$ 2 cut positions ($c = 0/c = 50$). Figures on the right showcase the marginal distribution *difference* between them and the ground-truth. It is noticeable from the figures that GPT did not learn cfg3g and cfg3f well. This is consistent with the generation accuracies in Figure 4.

# E  MORE EXPERIMENTS ON NT ANCESTOR AND NT BOUNDARY PREDICTIONS

## E.1  NT ANCESTOR AND NT BOUNDARY PREDICTIONS

Earlier, as confirmed in Figure 5, we established that the hidden states (of the final transformer layer) have implicitly encoded the NT ancestor symbols $\mathfrak{s}_\ell(i)$ for each CFG level $\ell$ and token position $i$ using a linear transformation. In Figure 18(a), we also demonstrated that the same conclusion applies to the NT-end boundary information $\mathfrak{b}_\ell(i)$. More importantly, for $\mathfrak{b}_\ell(i)$, we showed that this information is *stored locally*, very close to position $i$ (such as at $i \pm 1$). Detailed information can be found in Figure 18.

Furthermore, as recalled in Figure 9, we confirmed that at any NT boundary where $\mathfrak{b}_\ell(i) = 1$, the transformer has also locally encoded clear information about the NT ancestor symbol $\mathfrak{s}_\ell(i)$, either exactly at $i$ or at $i\pm1$. To be precise, this is a conditional statement — given that it is an NT boundary, NT ancestors can be predicted. Therefore, in principle, one must also verify that the prediction task for the NT boundary is successful to begin with. Such missing experiments are, in fact, included in Figure 18(b) and Figure 18(c).

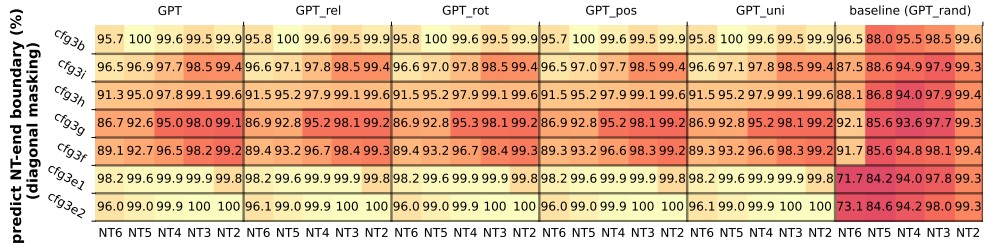

(a) Predicting NT boundaries: the column $NT_\ell$ for $\ell = 2, 3, 4, 5, 6$ represents the accuracy of predicting $\mathfrak{b}_\ell$ using the multi-head linear probing function described in (4.2).

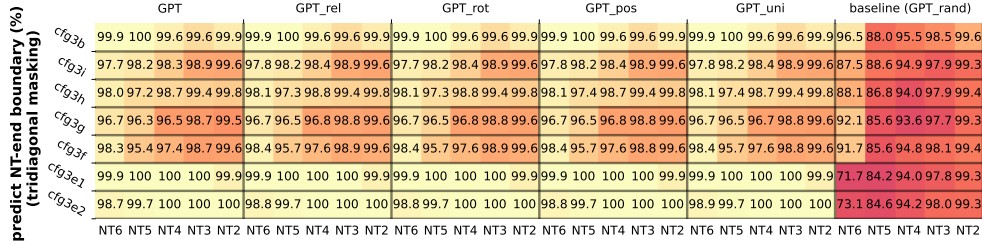

(b) Predicting NT boundaries with diagonal masking: the column $NT_\ell$ for $\ell = 2, 3, 4, 5, 6$ represents the accuracy of predicting $\mathfrak{b}_\ell$ using (4.2) but setting $w_{r, i \to k} = 0$ for $i \neq k$.

(c) Predicting NT boundaries with tridiagonal masking: the column $NT_\ell$ for $\ell = 2, 3, 4, 5, 6$ represents the accuracy of predicting $\mathfrak{b}_\ell$ using (4.2) but setting $w_{r, i \to k} = 0$ for $|i - k| > 1$.

Figure 18: After pre-training, the NT-end boundary information — i.e., $\mathfrak{b}_\ell(i)$ for position $i$ and NT level $\ell$ — is largely stored *locally* near the hidden state at position $i \pm 1$, up to a linear transformation. This can be compared with the prediction accuracy of the NT ancestor $\mathfrak{s}_\ell(i)$ in Figure 5.

**Observation.** This implies, the transformer actually *knows*, with a very good accuracy, that "position $i$ is already the end of NT on level $\ell$", by just reading all the texts until this position (possibly peeking one more to its right).

**Remark 1.** It may be mathematically necessary to peek more than 1 tokens to decide if a position $i$ is at an NT boundary, due to CFG's ambiguity. But, in most cases, that can be decided quite early.

**Remark 2.** Predicting NT boundary is a very *biased* binary classification task. For levels $\ell$ that are close to the CFG root, most symbols are not at NT boundary for that level $\ell$ (see Figure 2). For such reason, in the *heatmap color* of the figures above, we have *normalized* the columns with respect to NT2..NT6 differently, to reflect this bias.

### E.2 NT PREDICTIONS ACROSS TRANSFORMER'S LAYERS

As one may image, the NT ancestor and boundary information for smaller CFG levels $\ell$ (i.e., closer to CFG root) are only learned at those deeper transformer layers $l$. In Figure 19, we present this finding by calculating the *linear* encoding accuracies with respect to all the 12 transformer layers in GPT and $\text{GPT}_{\text{rel}}$. We confirm that generative models discover such information *hierarchically*.

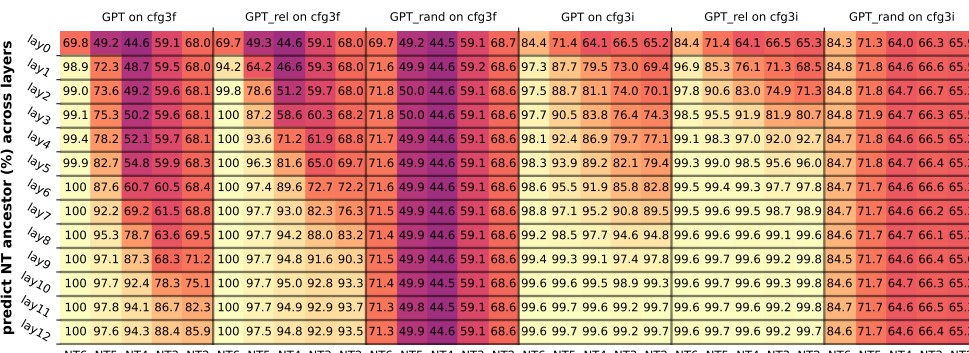

(a) Predict NT ancestors, comparing against the GPT_rand baseline

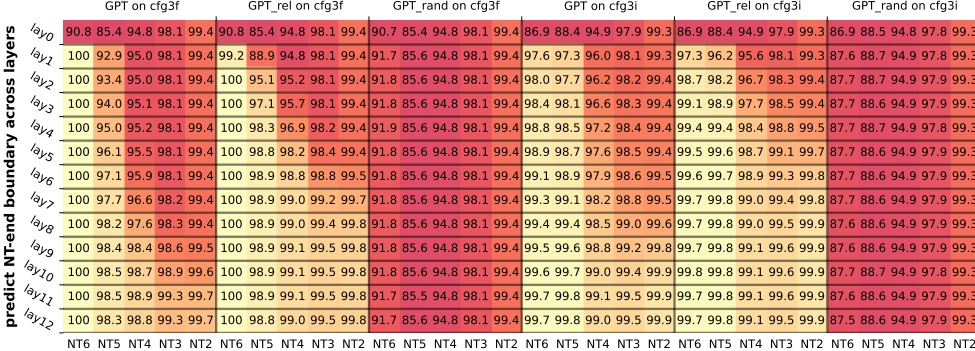

(b) Predict NT boundaries, comparing against the GPT_rand baseline

Figure 19: Generative models discover NT ancestors and NT boundaries hierarchically.

### E.3 NT Predictions Across Training Epochs

Moreover, one may conjecture that the NT ancestor and NT boundary information is learned *gradually* as the number of training steps increase. We have confirmed this in Figure 20. We emphasize that this does not imply layer-wise training is applicable in learning deep CFGs. It is crucial to train all the layers together, as the training process of deeper transformer layers may help backward correct the features learned in the lower layers, through a process called "backward feature correction" (Allen-Zhu & Li, 2023).

predict NT ancestor/boundary (%) across training epochs

| epoch | predict NT (GPT) | | | | | predict NTend (GPT) | | | | | predict NT (GPT_rel) | | | | | predict NTend (GPT_rel) | | | | |
|---|---|---|---|---|---|---|---|---|---|---|---|---|---|---|---|---|---|---|---|---|
| | NT6 | NT5 | NT4 | NT3 | NT2 | NT6 | NT5 | NT4 | NT3 | NT2 | NT6 | NT5 | NT4 | NT3 | NT2 | NT6 | NT5 | NT4 | NT3 | NT2 |
| 5 | 99.5 | 84.2 | 57.2 | 59.9 | 68.7 | 100 | 96.4 | 95.6 | 98.1 | 99.4 | 100 | 96.2 | 86.8 | 68.8 | 70.9 | 100 | 98.5 | 98.5 | 98.7 | 99.5 |
| 10 | 100 | 93.2 | 71.6 | 62.0 | 69.1 | 100 | 98.0 | 97.2 | 98.2 | 99.4 | 100 | 96.8 | 91.7 | 79.7 | 75.5 | 100 | 98.6 | 98.8 | 99.1 | 99.6 |
| 15 | 100 | 95.2 | 79.7 | 64.5 | 69.9 | 100 | 98.2 | 97.9 | 98.4 | 99.4 | 100 | 97.0 | 92.7 | 85.3 | 80.0 | 100 | 98.6 | 98.8 | 99.3 | 99.7 |
| 20 | 100 | 96.1 | 83.4 | 66.1 | 70.3 | 100 | 98.4 | 98.3 | 98.5 | 99.4 | 100 | 97.1 | 93.2 | 87.5 | 83.4 | 100 | 98.7 | 98.9 | 99.4 | 99.7 |
| 25 | 100 | 96.5 | 86.0 | 68.7 | 71.1 | 100 | 98.4 | 98.4 | 98.6 | 99.5 | 100 | 97.2 | 93.6 | 88.9 | 86.0 | 100 | 98.7 | 98.9 | 99.4 | 99.8 |
| 30 | 100 | 96.8 | 87.5 | 70.5 | 71.7 | 100 | 98.4 | 98.5 | 98.7 | 99.5 | 100 | 97.2 | 93.7 | 89.7 | 87.8 | 100 | 98.7 | 98.9 | 99.4 | 99.8 |
| 35 | 100 | 97.0 | 88.5 | 71.9 | 72.6 | 100 | 98.4 | 98.5 | 98.8 | 99.5 | 100 | 97.4 | 94.1 | 90.6 | 89.3 | 100 | 98.7 | 98.9 | 99.4 | 99.8 |
| 40 | 100 | 97.1 | 89.4 | 73.3 | 73.1 | 100 | 98.5 | 98.6 | 98.8 | 99.5 | 100 | 97.3 | 94.0 | 90.8 | 90.1 | 100 | 98.7 | 98.9 | 99.4 | 99.8 |
| 45 | 100 | 97.1 | 90.1 | 74.7 | 73.9 | 100 | 98.4 | 98.6 | 98.9 | 99.5 | 100 | 97.4 | 94.0 | 91.1 | 91.0 | 100 | 98.7 | 98.9 | 99.4 | 99.8 |
| 50 | 100 | 97.2 | 90.6 | 76.3 | 74.4 | 100 | 98.5 | 98.6 | 98.9 | 99.6 | 100 | 97.4 | 94.1 | 91.3 | 91.4 | 100 | 98.7 | 98.9 | 99.4 | 99.8 |
| 55 | 100 | 97.3 | 91.0 | 77.6 | 75.0 | 100 | 98.4 | 98.7 | 99.0 | 99.6 | 100 | 97.4 | 94.2 | 91.5 | 91.7 | 100 | 98.7 | 99.0 | 99.5 | 99.8 |
| 60 | 100 | 97.2 | 91.4 | 78.8 | 76.0 | 100 | 98.4 | 98.7 | 99.0 | 99.6 | 100 | 97.3 | 94.3 | 91.6 | 91.8 | 100 | 98.8 | 99.0 | 99.5 | 99.8 |
| 65 | 100 | 97.3 | 91.8 | 79.8 | 76.9 | 100 | 98.4 | 98.7 | 99.0 | 99.6 | 100 | 97.4 | 94.3 | 91.7 | 92.0 | 100 | 98.7 | 99.0 | 99.5 | 99.8 |
| 70 | 100 | 97.4 | 92.1 | 80.5 | 77.2 | 100 | 98.4 | 98.7 | 99.0 | 99.6 | 100 | 97.5 | 94.4 | 91.7 | 92.3 | 100 | 98.8 | 99.0 | 99.5 | 99.8 |
| 75 | 100 | 97.4 | 92.4 | 81.2 | 77.9 | 100 | 98.4 | 98.7 | 99.1 | 99.6 | 100 | 97.4 | 94.3 | 91.8 | 92.5 | 100 | 98.8 | 99.0 | 99.5 | 99.8 |
| 80 | 100 | 97.5 | 92.7 | 82.2 | 78.5 | 100 | 98.4 | 98.7 | 99.1 | 99.6 | 100 | 97.5 | 94.4 | 91.9 | 92.5 | 100 | 98.8 | 99.0 | 99.5 | 99.8 |
| 85 | 100 | 97.3 | 92.7 | 82.6 | 79.1 | 100 | 98.3 | 98.7 | 99.1 | 99.6 | 100 | 97.5 | 94.5 | 92.1 | 92.5 | 100 | 98.8 | 99.0 | 99.5 | 99.8 |
| 90 | 100 | 97.5 | 92.9 | 83.3 | 79.3 | 100 | 98.4 | 98.7 | 99.1 | 99.7 | 100 | 97.5 | 94.5 | 92.1 | 92.5 | 100 | 98.8 | 99.0 | 99.5 | 99.8 |
| 95 | 100 | 97.5 | 93.0 | 83.9 | 80.3 | 100 | 98.4 | 98.7 | 99.1 | 99.7 | 100 | 97.4 | 94.4 | 92.2 | 93.0 | 100 | 98.7 | 99.0 | 99.5 | 99.8 |
| 100 | 100 | 97.5 | 93.3 | 84.4 | 80.5 | 100 | 98.4 | 98.7 | 99.2 | 99.7 | 100 | 97.5 | 94.5 | 92.3 | 93.0 | 100 | 98.8 | 99.0 | 99.5 | 99.8 |
| 105 | 100 | 97.5 | 93.3 | 84.7 | 80.8 | 100 | 98.4 | 98.8 | 99.2 | 99.7 | 100 | 97.5 | 94.5 | 92.3 | 93.0 | 100 | 98.8 | 99.0 | 99.5 | 99.8 |
| 110 | 100 | 97.5 | 93.3 | 85.0 | 81.6 | 100 | 98.3 | 98.7 | 99.2 | 99.7 | 100 | 97.5 | 94.5 | 92.2 | 92.9 | 100 | 98.7 | 99.0 | 99.5 | 99.8 |
| 115 | 100 | 97.5 | 93.4 | 85.3 | 81.5 | 100 | 98.4 | 98.8 | 99.2 | 99.7 | 100 | 97.4 | 94.4 | 92.2 | 92.8 | 100 | 98.8 | 99.0 | 99.5 | 99.8 |
| 120 | 100 | 97.6 | 93.5 | 85.6 | 82.4 | 100 | 98.4 | 98.8 | 99.2 | 99.7 | 100 | 97.5 | 94.5 | 92.2 | 92.9 | 100 | 98.8 | 99.0 | 99.5 | 99.8 |
| 125 | 100 | 97.6 | 93.8 | 86.2 | 82.8 | 100 | 98.4 | 98.8 | 99.2 | 99.7 | 100 | 97.6 | 94.8 | 92.6 | 93.3 | 100 | 98.8 | 99.0 | 99.5 | 99.8 |
| 130 | 100 | 97.5 | 93.7 | 86.4 | 83.1 | 100 | 98.4 | 98.7 | 99.2 | 99.7 | 100 | 97.4 | 94.6 | 92.6 | 93.1 | 100 | 98.7 | 99.0 | 99.5 | 99.8 |
| 135 | 100 | 97.6 | 93.8 | 86.7 | 83.3 | 100 | 98.4 | 98.8 | 99.2 | 99.7 | 100 | 97.5 | 94.7 | 92.4 | 93.1 | 100 | 98.7 | 99.0 | 99.5 | 99.8 |
| 140 | 100 | 97.5 | 93.6 | 86.5 | 83.6 | 100 | 98.3 | 98.8 | 99.2 | 99.7 | 100 | 97.5 | 94.6 | 92.6 | 93.3 | 100 | 98.7 | 99.0 | 99.5 | 99.8 |
| 145 | 100 | 97.6 | 93.8 | 86.7 | 83.5 | 100 | 98.4 | 98.8 | 99.2 | 99.7 | 100 | 97.5 | 94.7 | 92.9 | 93.4 | 100 | 98.7 | 99.0 | 99.5 | 99.8 |
| 150 | 100 | 97.6 | 93.8 | 87.0 | 83.8 | 100 | 98.4 | 98.8 | 99.2 | 99.7 | 100 | 97.5 | 94.7 | 92.7 | 93.4 | 100 | 98.8 | 99.0 | 99.5 | 99.8 |
| 155 | 100 | 97.6 | 93.9 | 87.1 | 84.7 | 100 | 98.4 | 98.8 | 99.2 | 99.7 | 100 | 97.5 | 94.6 | 92.5 | 93.0 | 100 | 98.8 | 99.0 | 99.5 | 99.8 |
| 160 | 100 | 97.6 | 94.0 | 87.1 | 84.5 | 100 | 98.4 | 98.8 | 99.3 | 99.7 | 100 | 97.6 | 94.7 | 92.5 | 93.0 | 100 | 98.8 | 99.0 | 99.5 | 99.8 |
| 165 | 100 | 97.6 | 94.0 | 87.8 | 85.0 | 100 | 98.4 | 98.8 | 99.3 | 99.7 | 100 | 97.5 | 94.6 | 92.7 | 93.3 | 100 | 98.8 | 99.0 | 99.5 | 99.8 |
| 170 | 100 | 97.5 | 94.1 | 87.8 | 85.3 | 100 | 98.4 | 98.8 | 99.3 | 99.7 | 100 | 97.4 | 94.7 | 92.8 | 93.5 | 100 | 98.7 | 99.0 | 99.5 | 99.8 |
| 175 | 100 | 97.6 | 94.1 | 87.9 | 85.4 | 100 | 98.4 | 98.8 | 99.3 | 99.7 | 100 | 97.5 | 94.7 | 92.6 | 93.2 | 100 | 98.8 | 99.0 | 99.5 | 99.8 |
| 180 | 100 | 97.6 | 94.1 | 87.9 | 85.3 | 100 | 98.4 | 98.8 | 99.3 | 99.7 | 100 | 97.6 | 94.7 | 92.5 | 93.2 | 100 | 98.8 | 99.0 | 99.5 | 99.8 |
| 185 | 100 | 97.6 | 94.2 | 88.1 | 85.5 | 100 | 98.3 | 98.8 | 99.3 | 99.7 | 100 | 97.5 | 94.7 | 92.7 | 93.4 | 100 | 98.8 | 99.0 | 99.5 | 99.8 |
| 190 | 100 | 97.6 | 94.3 | 88.2 | 85.6 | 100 | 98.4 | 98.8 | 99.3 | 99.7 | 100 | 97.5 | 94.8 | 92.8 | 93.6 | 100 | 98.8 | 99.0 | 99.5 | 99.8 |
| 195 | 100 | 97.6 | 94.2 | 88.3 | 86.0 | 100 | 98.4 | 98.8 | 99.3 | 99.7 | 100 | 97.5 | 94.8 | 92.8 | 93.5 | 100 | 98.8 | 99.0 | 99.5 | 99.8 |
| 200 | 100 | 97.7 | 94.2 | 88.2 | 85.7 | 100 | 98.4 | 98.8 | 99.3 | 99.7 | 100 | 97.5 | 94.7 | 92.7 | 93.3 | 100 | 98.8 | 99.0 | 99.5 | 99.8 |

Figure 20: Generative models discover NT ancestors and NT boundaries gradually across training epochs (here 1 epoch equals 500 training steps). CFG levels closer to the leaves are learned faster, and their accuracies continue to increase as deeper levels are being learned, following a principle called "backward feature correction" in deep hierarchical learning (Allen-Zhu & Li, 2023).

# F    MORE EXPERIMENTS ON ATTENTION PATTERNS

## F.1    POSITION-BASED ATTENTION PATTERN

Recall from Section 5.1 that we asserted the transformer's attention weights are primarily influenced by the relative distance of the tokens. This remains true even when *trained on the CFG data* with *absolute positional embedding*. We omitted the details in the main body due to space constraints, but we will provide them now.

Formally, let $A_{l,h,j\to i}(x)$ for $j \geq i$ represent the attention weight for positions $j \to i$ at layer $l$ and head $h$ of the transformer, on input sequence $x$. For each layer $l$, head $h$, and distance $p \geq 0$, we compute the average of the partial sum $\sum_{1 \leq i' \leq i} A_{l,h,j\to i'}(x)$ over all data $x$ and pairs $i, j$ with $j - i = p$. We observe a strong correlation between the attention pattern and the relative distance $p = j - i$. The attention pattern is also *multi-scale*, with some attention heads focusing on shorter distances and others on longer ones. We plot this cumulative sum for different $l, h, p$ in Figure 21 in both $\mathrm{GPT}/\mathrm{GPT}_{\mathrm{rel}}$ for various datasets.

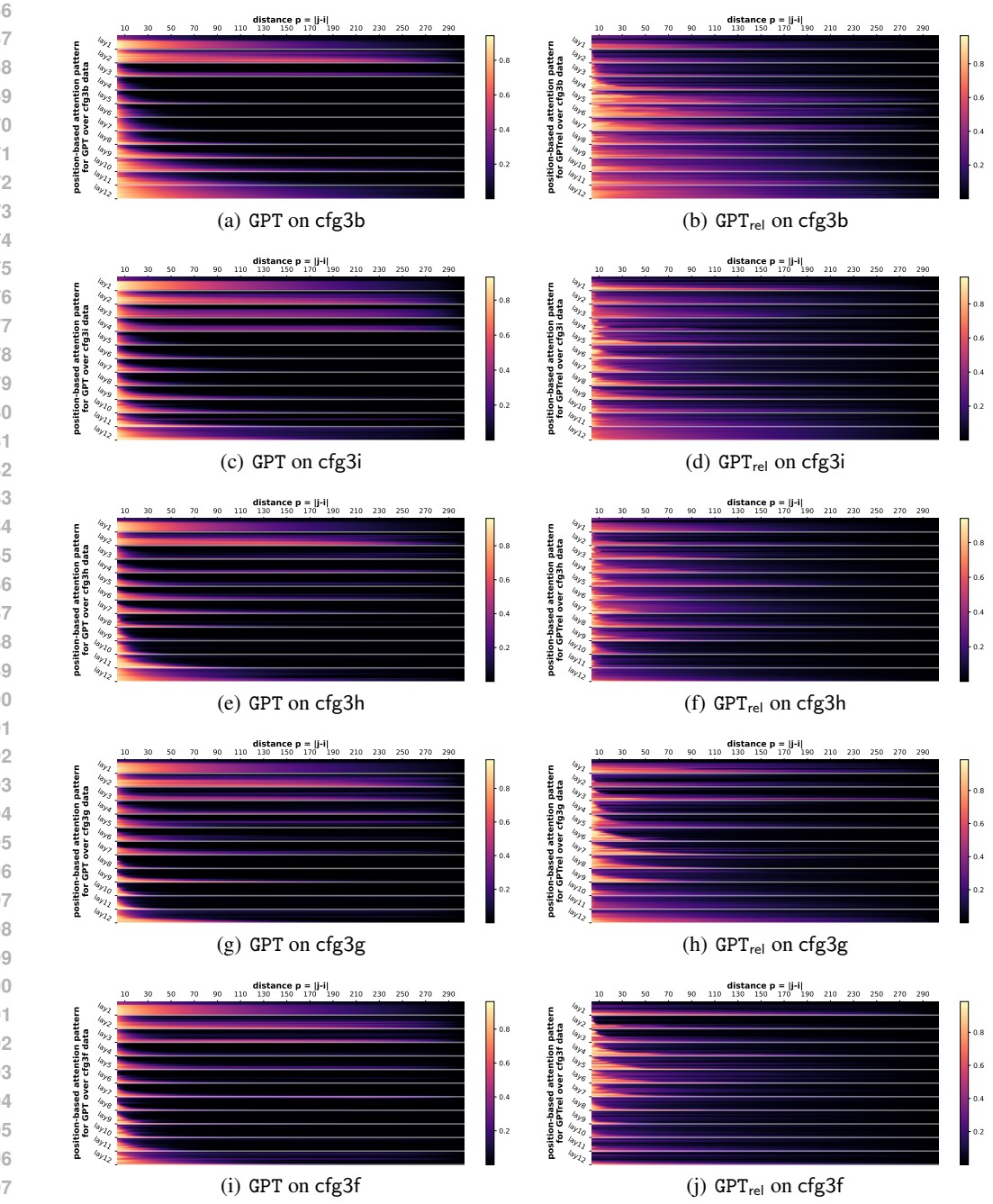

Figure 21: Position-based attention pattern. The 12 rows in each layer represent 12 heads. **Observations.** The attention pattern is multi-scale: different heads or layers have different dependencies on $p$.

## F.2 FROM ANYWHERE TO NT-ENDS

Recall from Figure 7(a), we showed that after removing the position-bias $B_{l,h,j\to i}(x) := A_{l,h,j\to i}(x) - \overline{A}_{l,h,j-i}$, the attention weights have a very strong bias towards *tokens $i$ that are at NT ends*. In Figure 22 we complement this experiment with more datasets.

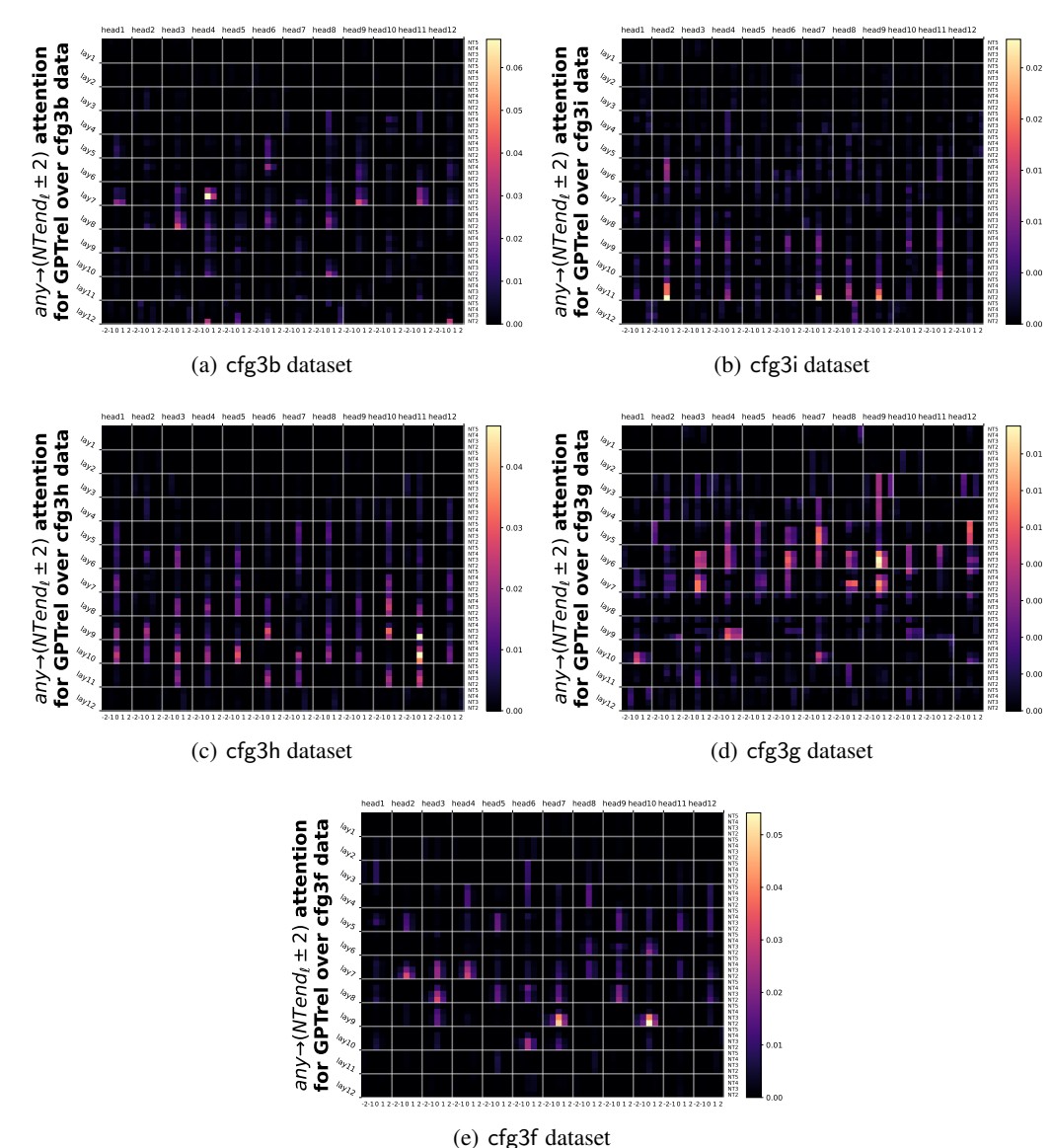

(a) cfg3b dataset

(b) cfg3i dataset

(c) cfg3h dataset

(d) cfg3g dataset

(e) cfg3f dataset

Figure 22: Attention weights $B_{l,h,j\to i}(x)$ averaged over data $x$ and pairs $i, j$ such that $i + \delta$ is at the NT-end in level $\ell$ of the CFG. In each cell, the four rows correspond to levels $\ell = 2, 3, 4, 5$, and the five columns represent $\delta = -2, -1, 0, +1, +2$.

**Observation.** Attention is largest when $\delta = 0$ and drops rapidly to the surrounding tokens of $i$.

## F.3 FROM NT-ENDS TO NT-ENDS

As mentioned in Section 5.2 and Figure 7(b), not only do tokens generally attend more to NT-ends, but among those attentions, *NT-ends* are also *more likely* to attend to NT-ends. We include this full experiment in Figure 23 for every different level $\ell = 2, 3, 4, 5$, between any two pairs $j \to i$ that are both at NT-ends for level $\ell$, for the cfg3 datasets.

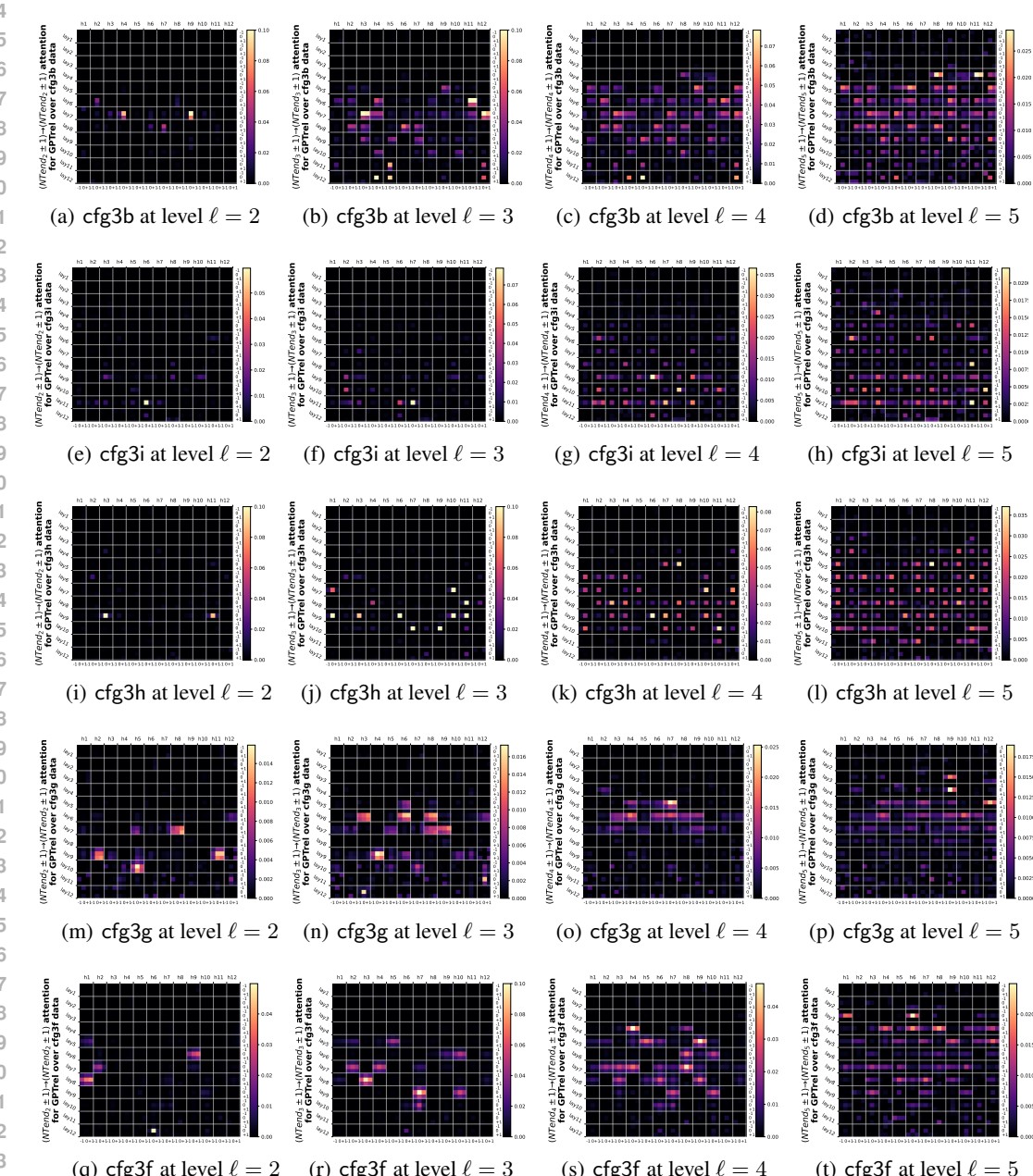

Figure 23: Attention pattern $B_{l,h,j\to i}(x)$ averaged over data $x$ and pairs $i, j$ such that $i + \delta_1$ and $j + \delta_2$ are at the NT-end boundaries in level $\ell$ of the CFG. In each block, the three rows correspond to $\delta_1 = -1, 0, +1$ and the three columns correspond to $\delta_2 = -1, 0, +1$.

**Observation.** Different transformer layer/head may be in charge of attending NT-ends at different levels $\ell$. Also, it is noticeable that the attention value drops rapidly from $\delta_1 = \pm 1$ to $\delta_1 = 0$, but less so from $\delta_2 = \pm 1$ to $\delta_2 = 0$. This should not be surprising, as it may still be ambiguous to decide if position $j$ is at NT-end *until* one reads few more tokens (see discussions under Figure 18).

## F.4  FROM NT-ENDS TO ADJACENT NT-ENDS

In Figure 7(c) we have showcased that $B_{l,h,j\to i}(x)$ has a strong bias towards *token pairs $i, j$ that are "adjacent" NT-ends*. We have defined what "adjacency" means in Section 5.2 and introduced a notion $B_{l,h,\ell'\to\ell,r}^{\text{end}\to\text{end}}$, to capture $B_{l,h,j\to i}(x)$ averaged over samples $x$ and all token pairs $i, j$ such that, they are at deepest NT-ends on levels $\ell, \ell'$ respectively (in symbols, $\flat^\sharp(i) = \ell \wedge \flat^\sharp(j) = \ell'$), and of

distance $r$ based on the ancestor indices at level $\ell$ (in symbols, $\mathfrak{p}_\ell(j) - \mathfrak{p}_\ell(i) = r$).

Previously, we have only presented by Figure 7(c) for a single dataset, and averaged over all the transformer layers. In the full experiment Figure 24 we show that for more datasets, and Figure 25 we show that for individual layers.

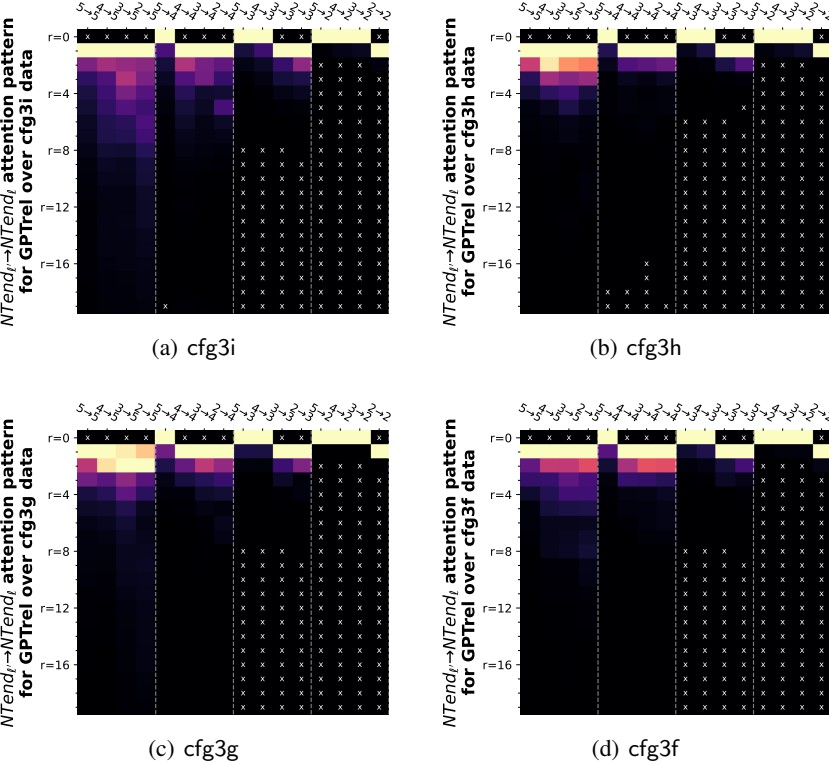

Figure 24: Attention pattern $B_{l,h,\ell'\to\ell,r}^{\text{end}\to\text{end}}(x)$ averaged over layers $l$, heads $h$ and data $x$. The columns represent $\ell' \to \ell$ and the rows represent $r$. "$\times$" means empty entries.

**Remark.** We present this boundary bias by looking at how close NT boundaries at level $\ell'$ attend to any other NT boundary at level $\ell$. For some distances $r$, this "distance" that we have defined may be non-existing. For instance, when $\ell \geq \ell'$ one must have $r > 0$. Nevertheless, we see that the attention value, *even after removing the position bias*, still have a large correlation with respect to the smallest possible distance $r$, between every pairs of NT levels $\ell, \ell'$. This is a strong evidence that CFGs are implementing some variant of dynamic programming.

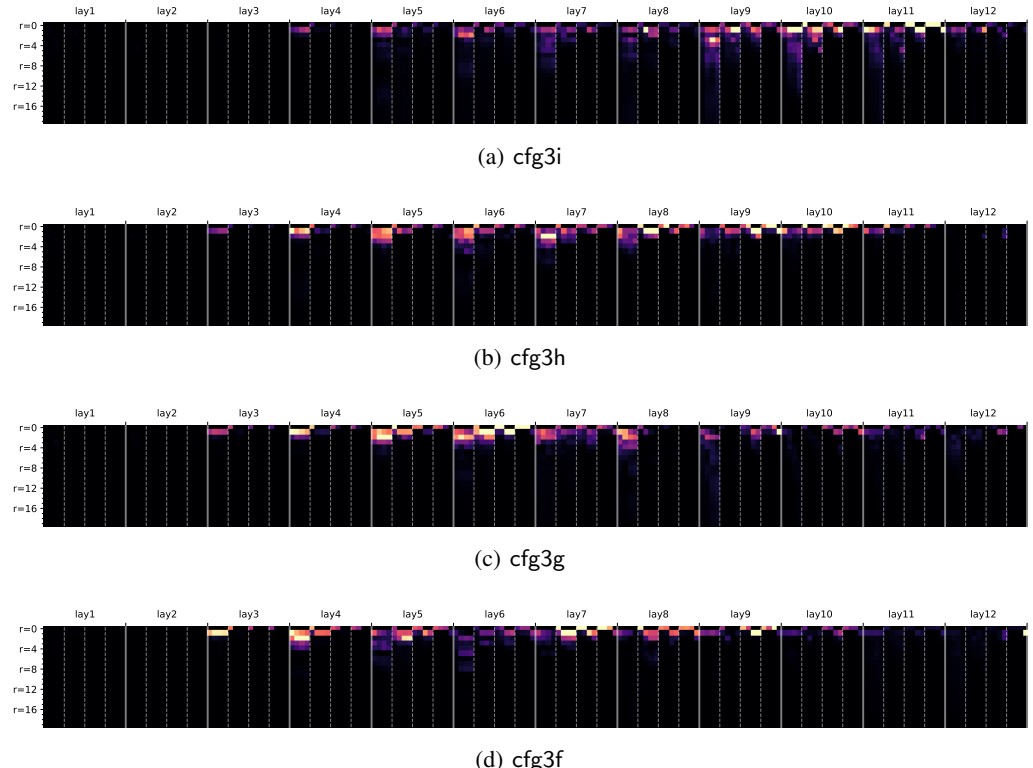

(a) cfg3i

(b) cfg3h

(c) cfg3g

(d) cfg3f

Figure 25: Attention pattern $B_{l,h,\ell'\to\ell,r}^{\text{end}\to\text{end}}(x)$ for each individual transformer layer $l \in [12]$, averaged over heads $h$ and data $x$. The rows and columns are in the same format as Figure 24.

**Observation.** Different transformer layers are responsible for learning "NT-end to most adjacent NT-end" at different CFG levels.

# G   MORE EXPERIMENTS ON IMPLICT CFGS

We study implicit CFGs where each terminal symbol $t \in \mathbf{T}$ is is associated a bag of observable tokens $\mathbf{OT}_t$. For this task, we study eight different variants of implicit CFGs, all converted from the exact same cfg3i dataset (see Section C.1). Recall cfg3i has three terminal symbols $|\mathbf{T}| = 3$:

- we consider a vocabulary size $|\mathbf{OT}| = 90$ or $|\mathbf{OT}| = 300$;
- we let $\{\mathbf{OT}_t\}_{t \in \mathbf{T}}$ be either disjoint or overlapping; and
- we let the distribution over $\mathbf{OT}_t$ be either uniform or non-uniform.

We present the generation accuracies of learning such implicit CFGs with respect to different model architectures in Figure 26, where in each cell we evaluate accuracy using 2000 generation samples. We also present the correlation matrix of the word embedding layer in Figure 10 for the $\mathrm{GPT}_\mathrm{rel}$ model (the correlation will be similar if we use other models).

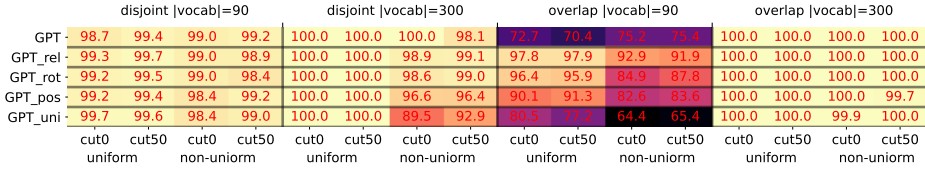

Figure 26: Generation accuracies on eight implicit CFG variants from pre-trained language models.

## H MORE EXPERIMENTS ON ROBUSTNESS

Recall that in Figure 11, we have compared clean training vs training over three types of perturbed data, for their generation accuracies given both clean prefixes and corrupted prefixes. We now include more experiments with respect to more datasets in Figure 27. For each entry of the figure, we have generated 2000 samples to evaluate the generation accuracy.

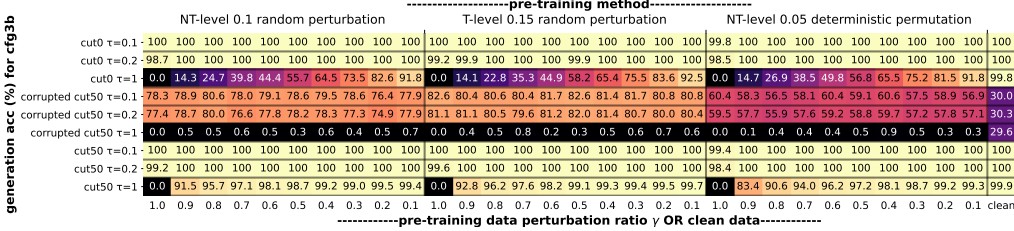

(a) cfg3b dataset

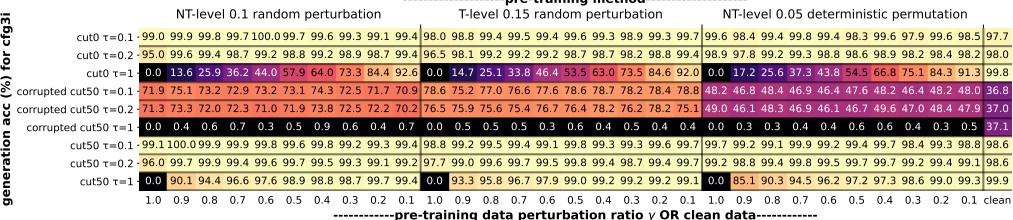

(b) cfg3i dataset

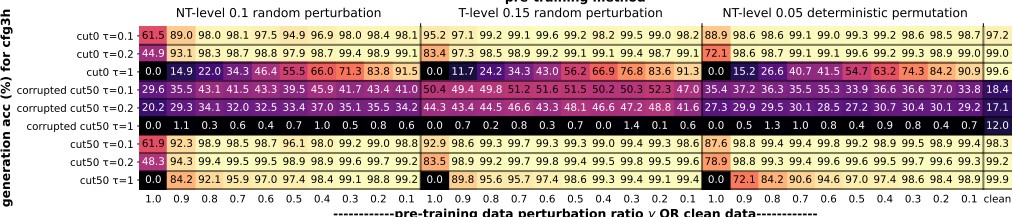

(c) cfg3h dataset

Figure 27: Generation accuracies for models pre-trained cleanly VS pre-trained over perturbed data, on clean or corrupted prefixes with cuts $c = 0$ or $c = 50$, using generation temperatures $\tau = 0.1, 0.2, 1.0$.

---

**Observation 1.** In Rows 4/5, by comparing against the last column, we see it is *beneficial* to include low-quality data (e.g. grammar mistakes) during pre-training. The amount of low-quality data could be little ($\gamma = 0.1$ fraction) or large (*every training sentence may have grammar mistake*).

**Observation 2.** In Rows 3/6/9 of Figure 11 we see pre-training teaches the model a *mode switch*. When given a correct prefix it is in the *correct mode* and completes with correct strings (Row 9); given corrupted prefixes it *always* completes sentences with grammar mistakes (Row 6); given no prefix it generates corrupted strings with probability $\gamma$ (Row 3).

**Observation 3.** Comparing Rows 4/5 to Row 6 in Figure 11 we see that high robust accuracy is achieved only when generating using low temperatures $\tau$. Using low temperature encourages the model to, for each next token, pick a more probable solution. This allows it to achieve good robust accuracy *even when* the model is trained totally on corrupted data ($\gamma = 1.0$).

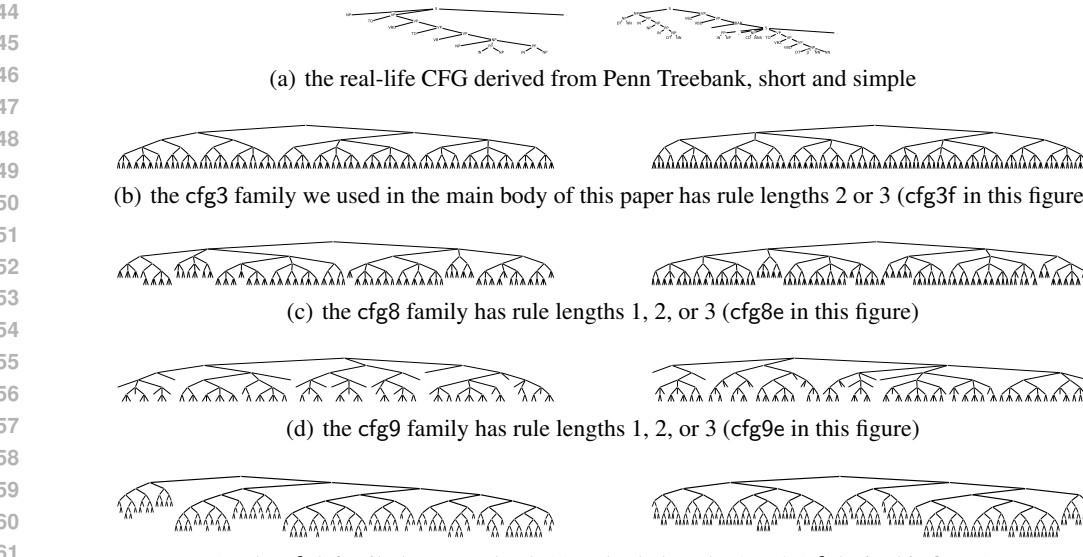

(a) the real-life CFG derived from Penn Treebank, short and simple

(b) the cfg3 family we used in the main body of this paper has rule lengths 2 or 3 (cfg3f in this figure)

(c) the cfg8 family has rule lengths 1, 2, or 3 (cfg8e in this figure)

(d) the cfg9 family has rule lengths 1, 2, or 3 (cfg9e in this figure)

(e) the cfg0 family has max-depth 11 and rule lengths 1 or 2 (cfg0e in this figure)

Figure 28: CFG comparisons: *left* is a medium-length sample and *right* is a 80%-percentile-length sample

# I  BEYOND THE CFG3 DATA FAMILY

The primary focus of this paper is on the cfg3 data family, introduced in Section C.1. This paper does not delve into how GPTs parse English or other natural languages. In fact, our CFGs are more "difficult" than, for instance, the English CFGs derived from the Penn TreeBank (PTB) (Marcus et al., 1993). By "difficult", we refer to the ease with which a human can parse them. For example, in the PTB CFG, if one encounters RB JJ or JJ PP consecutively, their parent must be ADJP. In contrast, given a string

```
33221312331211312321132231231211121321132231131132233312312112131133112132121333331232212131232221111213322131311311
31111113231233133133311331333332231211311121221111211233312331121113313333331123333131111333312113211312121133333321211
11212132232233221332211132211323233131111213223223221211133331121322221332211212133121331332212213221211121333123223312
```

that is in cfg3f, even with all the CFG rules provided, one would likely need a large piece of scratch paper to perform dynamic programming by hand to determine the CFG tree used to generate it.

Generally, the difficulty of CFGs scales with the average length of the strings. For instance, the average length of a CFG in our cfg3 family is over 200, whereas in the English Penn Treebank (PTB), it is only 28. However, the difficulty of CFGs may *inversely scale* with the number of Non-Terminal/Terminal (NT/T) symbols. Having an excess of NT/T symbols can simplify the parsing of the string using a greedy approach (recall the RB JJ or JJ PP examples mentioned earlier). This is why we minimized the number of NT/T symbols per level in our cfg3b, cfg3i, cfg3h, cfg3g, cfg3f construction. For comparison, we also considered cfg3e1, cfg3e2, which have many NT/T symbols per level. Figure 4 shows that such CFGs are extremely easy to learn.

To broaden the scope of this paper, we also briefly present results for some other CFGs. We include the *real-life* CFG derived from the Penn Treebank, and *three new families* of synthetic CFGs (cfg8, cfg9, cfg0). Examples from these are provided in Figure 28 to allow readers to quickly compare their difficulty levels.

## I.1  THE PENN TREEBANK CFG

We derive the English CFG from the Penn TreeBank (PTB) dataset (Marcus et al., 1993). To make our experiment run faster, we have removed all the CFG rules that have appeared fewer than 50 times in the data.[22] This results in 44 T+NT symbols and 156 CFG rules. The maximum node degree is

---

[22]These are a large set of rare rules, each appearing with a probability $\leq 0.2\%$. We are evaluating whether the generated sentence belongs to the CFG, a process that requires CPU-intensive dynamic programming. To make the computation time tractable, we remove the set of rare rules.

Note that cfg3 does not contain rare rules either. Including such rules complicates the CFG learning process, necessitating a larger transformer and extended training time. It also complicates the investigation of a

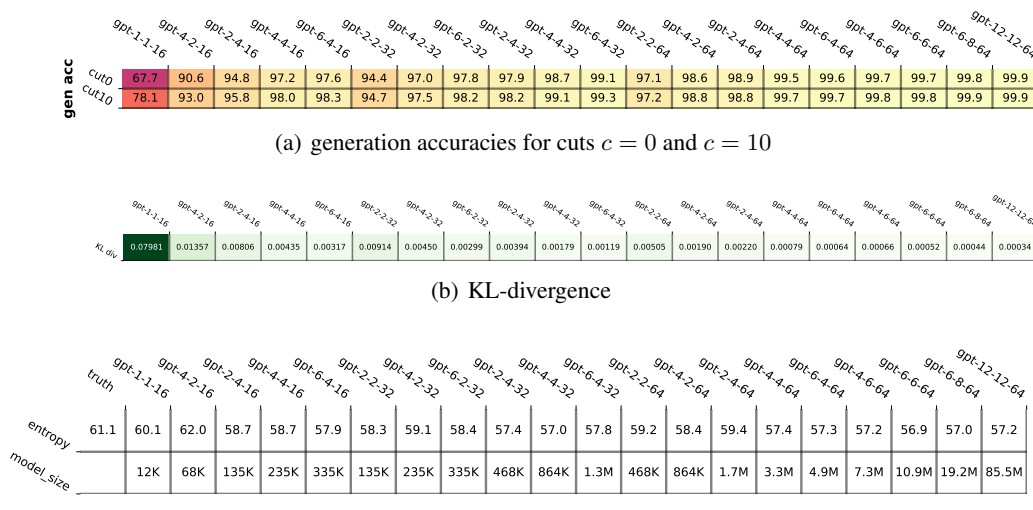

(a) generation accuracies for cuts $c = 0$ and $c = 10$

(b) KL-divergence

(c) entropy and model size

Figure 29: Real-life PTB CFG learned by $\texttt{GPT}_{\text{rot}}$ of different model sizes.

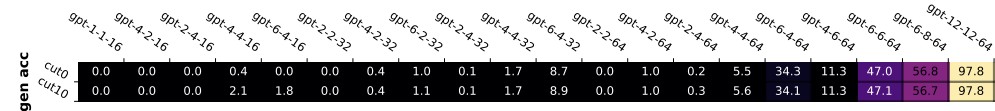

Figure 30: By contrast, small $\texttt{GPT}_{\text{rot}}$ model sizes cannot learn the cfg3f data (compare to Figure 29(a)).

65 (for the non-terminal NP) and the maximum CFG rule length is 7 (for S -> `` S , '' NP VP . ). If one performs binarization (to ensure all the CFG rules have a maximum length of 2), this results in 132 T+NT symbols and 288 rules.

*Remark* I.1. Following the notion of this paper, we treat those symbols such as NNS (common noun, plural), NN (common noun, singular) as *terminal symbols*. If one wishes to also take into consideration the bag of words (such as the word vocabulary of plural nouns), we have called it *implicit CFG* and studied it in Section B.1. In short, adding bag of words does not increase the learning difficult of a CFG; the (possibly overlapping) vocabulary words will be simply encoded in the embedding layer of a transformer.

For this PTB CFG, we also consider transformers of sizes *smaller* than GPT2-small. Recall GPT2-small has 12 layers, 12 heads, and 64 dimensions for each head. More generally, we let GPT-$\ell$-$h$-$d$ denote an $\ell$-layer, $h$-head, $d$-dim-per-head $\texttt{GPT}_{\text{rot}}$ (so GPT2-small can be written as GPT-12-12-64).

We use transformers of different sizes to pretrain on this PTB CFG. We repeat the experiments in Figure 4 (with the same pretrain parameters described in Appendix C.3), that is, we compute the generation accuracy, completion accuracy (with cut $c = 10$), the output entropy and the KL-divergence. We report the findings in Figure 29. In particular:

- Even a 135K-sized GPT2 (GPT-2-4-16) can achieve generation accuracy $\sim 95\%$ and have a KL divergence less than 0.01. (Note the PTB CFG has 30 terminal symbols so its KL divergence may appear larger than that of cfg3 in Figure 4.)

- Even a 1.3M-sized GPT2 (GPT-6-4-32) can achieve generation accuracy 99% and have a KL divergence on the order of 0.001.

- Using $M = 10000$ samples, we estimate the entropy of the ground truth PTB CFG is around 60 bits, and the output entropy of those learned transformer models are also on this magnitude.

- By contrast, those small model sizes cannot learn the cfg3f data, see Figure 30.

---

transformer's inner workings if these rare rules are not perfectly learned.

**cfg8 family (generation acc %)**

| | GPT | | GPT_rel | | GPT_rot | | GPT_pos | | GPT_uni | |
|---|---|---|---|---|---|---|---|---|---|---|
| | cut0 | cut20 | cut0 | cut20 | cut0 | cut20 | cut0 | cut20 | cut0 | cut20 |
| $cfg8_a$ | 99.6 | 99.6 | 99.9 | 99.9 | 99.9 | 99.9 | 99.9 | 99.9 | 99.9 | 99.8 |
| $cfg8_b$ | 99.8 | 99.8 | 100 | 100 | 100 | 100 | 100 | 100 | 99.9 | 99.9 |
| $cfg8_c$ | 95.3 | 95.2 | 99.4 | 99.4 | 99.2 | 99.2 | 98.7 | 98.6 | 98.8 | 98.8 |
| $cfg8_d$ | 97.5 | 97.5 | 98.3 | 98.3 | 98.0 | 98.0 | 97.9 | 97.9 | 97.6 | 97.4 |
| $cfg8_e$ | 82.1 | 82.3 | 97.4 | 97.6 | 93.7 | 93.7 | 94.6 | 94.4 | 93.0 | 93.5 |

**cfg9 family (generation acc %)**

| | GPT | | GPT_rel | | GPT_rot | | GPT_pos | | GPT_uni | |
|---|---|---|---|---|---|---|---|---|---|---|
| | cut0 | cut20 | cut0 | cut20 | cut0 | cut20 | cut0 | cut20 | cut0 | cut20 |
| $cfg9_a$ | 99.9 | 99.9 | 99.9 | 99.9 | 99.9 | 99.9 | 99.9 | 99.9 | 100 | 99.9 |
| $cfg9_b$ | 99.8 | 99.9 | 99.9 | 100 | 99.9 | 99.8 | 99.9 | 99.9 | 99.9 | 99.9 |
| $cfg9_c$ | 99.4 | 99.4 | 99.7 | 99.6 | 99.9 | 99.6 | 99.4 | 99.5 | 99.7 | 99.7 |
| $cfg9_d$ | 99.8 | 99.9 | 99.8 | 99.9 | 99.9 | 99.9 | 99.8 | 99.9 | 99.9 | 99.9 |
| $cfg9_e$ | 96.6 | 96.7 | 99.7 | 99.8 | 99.7 | 99.7 | 99.1 | 98.9 | 98.6 | 98.8 |

**cfg0 family (generation acc %)**

| | GPT | | GPT_rel | | GPT_rot | | GPT_pos | | GPT_uni | |
|---|---|---|---|---|---|---|---|---|---|---|
| | cut0 | cut20 | cut0 | cut20 | cut0 | cut20 | cut0 | cut20 | cut0 | cut20 |
| $cfg0_a$ | 97.4 | 97.5 | 98.9 | 98.8 | 98.3 | 98.4 | 98.5 | 98.5 | 98.5 | 98.4 |
| $cfg0_b$ | 90.9 | 91.3 | 96.0 | 95.9 | 94.1 | 93.1 | 92.9 | 92.8 | 92.5 | 92.5 |
| $cfg0_c$ | 99.5 | 99.6 | 99.6 | 99.7 | 99.6 | 99.6 | 99.7 | 99.7 | 99.6 | 99.6 |
| $cfg0_d$ | 98.0 | 98.3 | 98.5 | 98.6 | 98.4 | 98.5 | 98.7 | 98.8 | 98.1 | 98.2 |
| $cfg0_e$ | 99.7 | 99.8 | 99.7 | 99.7 | 99.7 | 99.7 | 99.7 | 99.8 | 99.7 | 99.7 |

Figure 31: Generation accuracies for cfg8/9/0 data family; suggesting our results *also hold for unbalanced trees with len-1 rules*.

## I.2 MORE SYNTHETIC CFGS

Remember that the cfg3 family appears "balanced" because all leaves are at the same depth and the non-terminal (NT) symbols at different levels are disjoint. This characteristic aids our investigation into the *inner workings* of a transformer learning such a language. We introduce three new synthetic data families, which we refer to as cfg8/9/0 (each with five datasets, totaling 15 datasets). These are all "unbalanced" CFGs, which support length-1 rules.[23] Specifically, the cfg0 family has a depth of 11 with rules of length 1 or 2, while the cfg8/9 family has depth 7 with rules of length 1/2/3. In all of these families, we demonstrate in Figure 31 that GPT can learn them with a satisfactory level of accuracy.

For this ICLR submission, we have included all the trees used in the supplementary materials. Below, we provide descriptions of how we selected them.

**CFG8 family.** The cfg8 family consists of five CFGs, namely cfg8a/b/c/d/e. They are constructed similarly to cfg3b/i/h/g/f, with the primary difference being that we sample rule lengths uniformly from $\{1, 2, 3\}$ instead of $\{2, 3\}$. Additionally,

- In cfg8a, we set the degree $|\mathcal{R}(a)| = 2$ for every NT $a$; we also ensure that in any generation rule, consecutive pairs of terminal/non-terminal symbols are distinct. The size is $(1, 3, 3, 3, 3, 3, 3)$.
- In cfg8b, we set $|\mathcal{R}(a)| = 2$ for every NT $a$; we remove the distinctness requirement to make the data more challenging than cfg8a. The size is $(1, 3, 3, 3, 3, 3, 3)$.
- In cfg8c, we set $|\mathcal{R}(a)| \in \{2, 3\}$ for every NT $a$ to make the data more challenging than cfg8b. The size is $(1, 3, 3, 3, 3, 3, 3)$.
- In cfg8d, we set $|\mathcal{R}(a)| = 3$ for every NT $a$. We change the size to $(1, 3, 3, 3, 3, 3, 4)$ because otherwise a random string would be too close (in editing distance) to this language.
- In cfg8e, we set $|\mathcal{R}(a)| \in \{3, 4\}$ for every NT $a$. We change the size to $(1, 3, 3, 3, 3, 3, 4)$ because otherwise a random string would be too close to this language.

A notable feature of this data family is that, due to the introduction of length-1 rules, a string in this language $L(\mathcal{G})$ may be *globally ambiguous*. This means that there can be multiple ways to parse it by the same CFG, resulting in multiple solutions for its NT ancestor/boundary information *for most symbols*. Therefore, it is not meaningful to perform linear probing on this dataset, as the per-symbol NT information is mostly non-unique.[24]

**CFG9 family.** Given the ambiguity issues arising from the cfg8 data construction, our goal is to construct an unbalanced and yet challenging CFG data family where the non-terminal (NT) information is mostly unique, thereby enabling linear probing.

To accomplish this, we first adjust the size to $(1, 4, 4, 4, 4, 4, 4)$, then we permit only one NT per layer to have a rule of length 1. We construct five CFGs, denoted as cfg9a/b/c/d/e, and their degree configurations (i.e., $\mathcal{R}(a)$) are identical to those of the cfg8 family. We then employ rejection sampling by generating a few strings from these CFGs and checking if the dynamic programming (DP) solution is unique. If it is not, we continue to generate a new CFG until this condition is met.

Examples from cfg9e are illustrated in Figure 28. We will conduct linear probing experiments on this data family.

---

[23]When a length-1 CFG rule is applied, we can merge the two nodes at different levels, resulting in an "unbalanced" CFG.

[24]In contrast, the cfg3 data family is only *locally* ambiguous, meaning that it is difficult to determine its hidden NT information by locally examining a substring; however, when looking at the entire string as a whole, the NT information per symbol can be uniquely determined with a high probability (if using for instance dynamic programming).

Figure 32: Same as Figure 5 but for the `cfg9` family. After pre-training, hidden states of generative models implicitly encode the NT ancestors information. The $NT_\ell$ column represents the accuracy of predicting $\mathfrak{s}_\ell$, the NT ancestors at level $\ell$. This suggests our probing technique applies more broadly.

Figure 33: Same as Figure 9 but for the `cfg9` data family. Generative pre-trained transformer encodes NT ancestors almost exactly _at_ NT boundaries. The $NT_\ell$ column represents the accuracy of predicting $\mathfrak{s}_\ell(i)$ at locations $i$ with $\mathfrak{b}_\ell(i) = 1$. This suggests our probing technique applies more broadly.

**CFG0 family.** Since all the CFGs above support rules of length 3, we have focused on $L = 7$ to prevent the string length from becoming excessively long.[25] In the cfg0 family, we construct five CFGs, denoted as cfg0a/b/c/d/e. All of them have a depth of $L = 11$. Their rule lengths are randomly selected from $\{1, 2\}$ (compared to $\{2, 3\}$ for cfg3 or $\{1, 2, 3\}$ for cfg8/9). Their degree configurations (i.e., $\mathcal{R}(a)$) are identical to those of the cfg8 family. We have chosen their sizes as follows, noting that we have enlarged the sizes as otherwise a random string would be too close to this language:

- We use size $[1, 2, 3, 4, 4, 4, 4, 4, 4, 4, 4]$ for cfg0a/b.
- We use size $[1, 2, 3, 4, 5, 6, 6, 6, 6, 6, 6]$ for cfg0c.
- We use size $[1, 2, 3, 4, 5, 6, 7, 8, 9, 10, 11]$ for cfg0d/e.

Once again, the CFGs generated in this manner are globally ambiguous like the cfg8 family, so we cannot perform linear probing on them. However, it would be interesting to demonstrate the ability of transformers to learn such CFGs.

**Additional experiments.** We present the generation accuracies (or the complete accuracies for cut $c = 20$) for the three new data families in Figure 31. It is evident that the cfg8/9/0 families can be learned almost perfectly by GPT2-small, especially the relative/rotary embedding ones.

As previously mentioned, the `cfg9` data family is not globally ambiguous, making it an excellent synthetic data set for testing the encoding of the NT ancestor/boundary information, similar to what we did in Section 4. Indeed, we replicated our probing experiments in Figure 32 and Figure 33 for the `cfg9` data family. This suggests that **our probing technique has broader applicability.**

---

[25]Naturally, a larger transformer would be capable of solving such CFG learning tasks when the string length exceeds 1000; we have briefly tested this and found it to be true. However, conducting comprehensive experiments of this length would be prohibitively expensive, so we have not included them in this paper.

