# OpenReview forum: "Interpretability of Language Models for Learning Hierarchical Structures"
_ICLR.cc/2025/Conference — Submitted to ICLR 2025_

### Official Review · Reviewer_4wx3 · 2024-10-22

**Soundness:** 3
**Presentation:** 2
**Contribution:** 3
**Rating:** 6
**Confidence:** 3

**Summary:**

This work investigates how LM grasp complex, recursive language structures defined by context-free grammars (CFGs). It develops a multi-head linear probing method to verify that the model’s hidden states linearly encode NT (non-terminal) information, which is interesting as pre-training on CFG-generated strings does not expose the CFG structure. The results also suggests that GPT models learn CFGs by implementing a dynamic programming-like algorithm.

**Strengths:**

(1) I think the idea of studying LM's understanding of synthetic CFG is super interesting and novel.

(2) Some of the findings (that LM can really learn those complicated CFG, and implementing DP-like operation inside) are quite surprising.

**Weaknesses:**

(1) The paper very condensed, and I would enjoy reading it more if it is simplified (including figures, tables, notations, etc.).

(2) The scope is centered around CFG, which is novel, but still limited.

**Questions:**

N/A.

---

> ### Author Response · Authors · 2024-11-28
> **Response to Reviewer 4wx3**
>
> We thank the review for the comments. As for the comments regarding the paper being very condensed, do you have some concrete suggestions regarding how we can improve the writing? It’s hard to please everyone, and this current paper format is after us communicating with ~20 readers and taking into account their taste of reading/writing styles.
>
>
> Also, many of the findings are not conventional, including our data, our probing method, so we tried to spend the main body explaining why we designed the experiments this way, and how exactly we certify that the GPT2 is learning to implement a DP. It becomes hard to remove any of these without confusing the readers.
>
>
> Of course, we'd love to hear more suggestions and wouldn't mind revising this paper further.  Thank you!

---

### Official Review · Reviewer_zggS · 2024-11-01

**Soundness:** 4
**Presentation:** 4
**Contribution:** 3
**Rating:** 8
**Confidence:** 3

**Summary:**

The paper proposes an analytical method for explaining the inner workings of Transformers. By defining CFG rules with specific properties to generate synthetic data for language models to learn, the consistency between the model's outputs and the CFG rules is observed using various methods, including generating subsequent content from a given prefix to observe whether it strictly follows the CFG rules, as well as probing the hidden states with methods like probing heads. Numerous pieces of evidence suggest that the model's hidden states can accurately capture the structural information of the CFG, while also verifying that self-attention works like dynamic programming.

**Strengths:**

1. Although the self-attention in Transformers has been speculated to simulate structures similar to human languages, and many related works have empirically validated this point (e.g. https://arxiv.org/abs/2004.14786), there have been few systematic demonstrations. This paper systematically demonstrates that Transformers can learn CFG grammars with long-range dependencies and local ambiguities, two properties that are highly relevant to human language. The entire deduction approach is scientific and rigorous.

2. The method of defining generative rules with certain properties to generate synthetic data and then simulating and analyzing it with Transformers may provide valuable insights for future related work.

3. The paper is well-written, with a clear and organized structure.

**Weaknesses:**

1. Considering programming languages, which are also CFGs, Transformers may not only learn the syntax of a program but also simulate the execution logic, such as sorting and other complex behaviors. These capabilities can also be learned by large models. Therefore, the inner workings of Transformers seem not limited to latent structure learning but also involve fitting causal logic. Learning CFGs may be a subset of their capabilities, though I acknowledge that this is not something that can be fully discussed in a single work.

**Questions:**

1. Some studies, such as (https://arxiv.org/abs/2203.00281), explicitly compose tokens via dynamic programming to learn latent structures and sentence representations. These methods even perform slightly better than Transformers on text representation tasks, yet their reasoning capabilities are significantly weaker. They can also learn CFG structures and even explicitly model the dynamic programming process, similar to the abilities of Transformers described in the paper. So what accounts for their disparity on reasoning ability ?

---

> ### Author Response · Authors · 2024-11-28
> **Response to Reviewer zggS**
>
> We sincerely thank the reviewer for supporting this work, and as you have pointed out, transformers work much much beyond fitting causal logic. We wish to use this study as a first step towards (more or less) understanding how LLMs learn “formats” or “structures” before we delve into more meaningful tasks, such as sorting or more complex forms of reasoning, etc.
> Also, a particular goal of this paper is to refute the argument that LM is a stochastic parrot, so clearly it is not relying on some simple (local) pattern recognition protocol to generate, but can rely on some more involved algorithm (such as DP).
>
>
> As for your question regarding works that “explicitly compose tokens via dynamic programming to learn latent structures and sentence representations”, we have two thoughts.
>
>
> A simple thought is that, adding task-specific designs (including a CKY-style encoder, or a p(si | s1..{i-1} s{i+1}..n) type of pretraining objective like 2203.00281), might make us a little farther away from the grand goal of AGI. We could be wrong, but a more desirable goal is to use some unified model / objective for all possible AI tasks whenever possible.
>
>
> Also, such MaskedLM type of objective (studied in 2203.00281) still makes me worry once again of how deep their method can go — we argued in this paper that BERT-like models fail to learn deep structures because of MaskedLM, and that that PennTree Bank (and possible most real-life language CFG) may not be very “deep”, and this might explain the success of  2203.00281. For a more “complex” CFG like ours, we are curious what’s their performance.
>
>
> A more important thought can connect to Chain of Thought (CoT). In this submission, we demonstrated that the DP process is learned implicitly. More generally (including our own follow-ups, not to cite due to anonymity), GPT2 can be good at learning other implicit tasks, without CoT. Whenever adding the intermediate steps (such as explicitly teaching the model to do DP), then the learning task may become easier.
>
>
> Ultimately, we perhaps want a delicate balance: we don’t want the AGI model to do A+B+C step by step; we also don’t expect it to solve a hard math problem without CoT. We think it is interesting to study what’s the balance here, in particular, what are the tasks that can be learned implicitly, in order to better understand what type of model/data we need in tomorrow’s large model training.
>
> We hope this to some extent answers your questions! Thanks again!

---

### Official Review · Reviewer_ura6 · 2024-11-01

**Soundness:** 3
**Presentation:** 1
**Contribution:** 3
**Rating:** 3
**Confidence:** 3

**Summary:**

This paper studies how well transformer language models can learn to process structured context-free languages, specifically those that require handling long-range dependencies and resolving local ambiguities. To do this, the authors use artificially generated context-free languages and find that transformers can learn these languages, but the type of positional encoding used in the model makes a difference. They also introduce a probing method to test how well the models learn the structure of a context-free language. The authors suggest that transformers might be processing input strings similarly to dynamic programming techniques used in context-free parsing.

**Strengths:**

The paper makes useful contributions in studying general language models with targeted formal languages. I find the following to be particularly useful:
- Examining language models with formal languages is a valuable approach for understanding their capabilities and limitations.
- The use of synthetic grammars is well thought out and effective, especially since these grammars are designed to test specific properties, like handling long dependencies.
- The results cover a good range of transformer design choices, providing a thorough look at how these models handle different CFG structures.

**Weaknesses:**

There are areas where clarity and focus of the paper could improve:
- **Complexity in Presentation**: The paper is sometimes hard to follow. The Introduction mixes motivation, literature review, and specific results, which makes it less effective as an overview. The amount of notation used and the naming of the CFGs, like `cfg3a`, is also hard to connect with anything concrete, and references to niche topics like the 2006 IOI problem add to the complexity.
- **Dense Results Section**: With so many results presented, organization is key to helping readers keep track. Referring to results by descriptive names rather than numbers might help readers better understand each finding at a glance.
- **Setup Comments**:
    - It’s worth noting that long-range dependencies don’t require context-free languages; even simple regular languages (e.g., “first symbol = last symbol”) can capture long dependencies.
    - The CFGs defined in the paper technically describe finite languages (and are therefore regular) because recursion depth is limited by L. This limitation might make it hard to tell whether the models are learning true recursion or just memorizing up to the required depth.
- **Comparison to Related Work**: Adding a discussion of related studies, especially those testing LMs on the Chomsky hierarchy, would help position the results more clearly within the existing research landscape.

**Questions:**

- In the Conclusion: While you might not want to discuss ongoing or future work in detail, it would be helpful to hear a bit more about potential directions or applications of this research.
- RNNs are often compared to finite-state automata or pushdown automata. Did you consider comparing transformers to simple RNNs or LSTMs in this study?

---

> ### Author Response · Authors · 2024-11-28
> **Response to Reviewer ura6**
>
> We sincerely thank the reviewer for the time raising the questions. Let us try to follow-up with your main comments.
>
> > It’s worth noting that long-range dependencies don’t require context-free languages; even simple regular languages (e.g., “first symbol = last symbol”) can capture long dependencies.
>
> **Our response:**
> That is correct. This is also the motivation of our paper, as mentioned on Line 48 to study “how language models perform **hard tasks, involving deep logics** / reasoning / computation chains”. We designed the CFG to be exactly not of such simpler form (like “first symbol = last symbol”).
>
>
> > The CFGs defined in the paper technically describe finite languages (and are therefore regular) because recursion depth is limited by L. This limitation might make it hard to tell whether the models are learning true recursion or just memorizing up to the required depth.
>
>
> **Our response:**
> Can we kindly ask the reviewer what you concretely mean by “memorizing up to the required depth”? If it’s memorizing the strings then we said in Line 181 that there are at least 10^140 possible strings (of our chosen depth L); the model can't even see all such possibilities, not to say memorize them. The model only has 10^8 parameters.
>
>
> If what you actually mean is that the model has memorized the small set of “rules”, then we believe this is not memorization but rather it has learned to parse/generate CFGs according to the right algorithm?
>
>
> > RNNs are often compared to finite-state automata or pushdown automata. Did you consider comparing transformers to simple RNNs or LSTMs in this study?
> >  In the Conclusion: While you might not want to discuss ongoing or future work in detail, it would be helpful to hear a bit more about potential directions or applications of this research.
>
>
> **Our response:**
> Thanks and not in this study. We choose decoder-only models due to their popularity and maturity in the AI community. Another very interesting candidate is Mamba, one of the most popular RNN-type of model, but there’s one concrete tradeoff issue. Mamba’s SSM layer has (3D)xD dimension (serving as long-term memory), where in full attention one can read from LxD dimension data (all the previous L tokens times hidden dimension D). This means, Mamba will be more desirable only when L>>3D. In this regime, the most interesting study IMHO would be whether Mamba can learn a very long CFG with small dimension D. This setup will be significantly different from this current submission, so we are **exploring this as a follow-up**.

---

### Official Review · Reviewer_Mj12 · 2024-11-04

**Soundness:** 4
**Presentation:** 3
**Contribution:** 4
**Rating:** 8
**Confidence:** 3

**Summary:**

This paper systematically investigates how Transformer language models learn and process complex recursive language structures by designing a set of synthetic context-free grammars (CFGs) with long-range dependencies and local ambiguities. The main contributions are: (1) introducing a multi-head linear probing method to analyze internal representations; (2) developing new visualization and quantification tools to understand attention patterns; (3) demonstrating that GPT models can learn complex CFGs in a dynamic programming-like manner and revealing the critical role of boundary-based attention in handling long-range dependencies; (4) validating the model's effectiveness and robustness in handling such complex language structures. These findings offer new insights into the inner workings of large language models.

**Strengths:**

The highlight of this paper is its novel approach to studying large language models by using carefully designed synthetic CFGs to explore model mechanisms. This method is innovative because it offers a controlled, quantifiable experimental environment, making complex model behaviors interpretable. By demonstrating how CFGs can be used to investigate the learning processes of models, this work provides a powerful analytical framework for future researchers. Combining theoretical analysis with experimental design, this approach not only aids in understanding existing models but also inspires new directions for probing the inner workings of even larger language models.

**Weaknesses:**

The paper mainly focuses on studying canonical CFGs, and the exploration of non-canonical CFGs is not sufficiently in-depth.

The research results are primarily based on specific synthetic CFGs, and their generalizability needs to be further verified.

**Questions:**

The paper suggests that Transformer models learn CFGs by emulating dynamic programming (DP) algorithms. Would it be possible to test this hypothesis more directly, perhaps by comparing the model’s internal representations to the states of known DP algorithms? The current evidence feels somewhat indirect.

The paper also notes BERT’s weaker performance in predicting deep non-terminal (NT) structures but doesn’t delve into the reasons behind this. While attributing it to the locality of the MLM task seems plausible, more direct evidence would strengthen this claim.

Could the authors discuss potential limitations? Acknowledging the method’s constraints would be helpful for future research directions.

---

> ### Author Response · Authors · 2024-11-28
> **Response to Reviewer Mj12**
>
> We sincerely thank the reviewer for supporting this work, and let us try to address your questions.
>
> **Question:**
> > The paper suggests that Transformer models learn CFGs by emulating dynamic programming (DP) algorithms. Would it be possible to test this hypothesis more directly, perhaps by comparing the model’s internal representations to the states of known DP algorithms? The current evidence feels somewhat indirect.
>
> **Answer:**
> The difficulty is that there are “infinitely” many ways to implement even a specific dynamic programming. While humans typically write code like “for k = i to j” and break whenever it suffices to certify that the subsequence from i to j can be generated from a CFG node, language models do this (kind of) in an arbitrary order. This is why we can only verify that those DP states in the backbone (i.e., DP states shared by essentially all DP algorithms, regardless of the implementation) are reached by the language model.
>
> **Question:**
> > The paper also notes BERT’s weaker performance in predicting deep non-terminal (NT) structures but doesn’t delve into the reasons behind this. While attributing it to the locality of the MLM task seems plausible, more direct evidence would strengthen this claim.
>
> **Answer:**
> Our original way of thinking is to contrast the NT prediction accuracies on lower/higher NTs to show that BERT “does not reason very deeply” on such CFG data. We’ll be more than happy to discuss with you regarding what you think can be “more direct evidence”.
>
> One proposal is to contrast two very close CFGs, one has root->a b b and the other uses root -> a a b, where all other CFG rules stay the same. In the most extreme case, let a = 11..1122..2211..11 and b = 11..1133..3311..11 where “11..11” means 100 copies of 1 and similarly for “22..22” and “33..33”. Then if you use masked LM then the BERT model will certainly learn 1 from its surrounding 1’s, and 2 from its surrounding 2’s, 3 from its surrounding 3’s. It will not distinguish root -> a b b vs root -> a a b.  (One can make this more complex by designing more complex / deeper a and b CFG nodes.)
>
> But how to test this? Unless you mask out the 100 consecutive 2’s or 3’s you cannot tell the difference; but once you do that, it’s not a typical MLM data (since in MLM, one randomly masks tokens out) and it’s not surprising that BERT models cannot perform well on this.
>
> Another proposal is to use BERT to do generation, to put all the tokens to the right as <MASK> and use the first <MASK> to do generation. This is in the same spirit as above, and the performance is poor as expected.
>
> Overall, in our mind, a “direct evidence” sounds like directly applying BERT models to see the difference, but this seems to require masking out a large consecutive number of tokens and demonstrate that BERT’s performance is poor. While this can be easily done, it does not seem convincing, and BERT is not trained on such type of data. — This also explains exactly why BERT prefers local reasoning.
>
> **Question:**
> > Could the authors discuss potential limitations? Acknowledging the method’s constraints would be helpful for future research directions.
>
> **Answer:**
> Certainly, for instance we designed the CFG to be “canonical” in order for the probing to better succeed. If there are global ambiguities then (meaning there’s no unique answer to the probing, such as if a->b c; b->1 2 or 1 2 3, c-> 3 1 or 1, then when a-> 1 2 3 1 there’s no unique answer to where this 3 comes from) then it will naturally fail.
>
> Also, we used “infinite” training data and the performance shall degrade if one controls the amount of training data. We intentionally left these out because we want to see, in a controlled environment (with infinite data + canonical CFG) what’s the capability of GPT2. We want to for instance refute the argument that LM is stochastic parrot, so clearly it is not relying on some simple (local) pattern recognition protocol to generate, but can rely on some more involved algorithm such as DP.
>
> Another technical limitation is our probing technique, where we did linear probing to extract the NT information, where the NT has just a dozen choices. When probing from dimension D networks, an information that has M possible choices, there is a linear head of dimension D x M. When the information we wish to probe is diverse (e.g., M>>100), this can be a problem — too many trainable params, etc. In our follow-up work we actually developed additional probing techniques to tackle situations like this, and let us not go into details to remain anonymous.
>
> Thanks again for your time!

---

### Meta-Review · Area_Chair_xWdx · 2024-12-20

**Metareview:**

The paper studies whether and how autoregressive Transformers are able to (implicitly) learn context-free grammars from being just trained on terminal tokens (i.e., without access to the underlying structure). It finds that Transformers are able to learn a variety of (simple) grammars, and probing/interpretability studies find evidence that this learning of the underlying grammar is due to the model actually learning the underlying generative model (rather than, for example, a lookup table).

On the positive side, some of the analyses are interesting, and the empirical experiments are generally well done. On the negative side, the model studies a very restricted/naive form a grammar (in particular, the grammar is not even recursive (!), which is a key feature underlying human language), and many of the analyses seems incomplete (for example, you could compare the next-word distributions from the actual grammar by using a probabilistic version of Earley's algorithm [1] to see how it matches the Transformer's next-word distribution, instead of the awkward calculation of the KL in result 3). Moreover, the writing was difficult to follow, and had weird snippets that seemed to suggest that this paper was intended to be submitted elsewhere (e.g., in section to the authors write "(for example, Chapter 3.1 should be only followed by Chapter 3.1.1, Chapter 4 or Chapter 3.2, not others)", but I don't see any chapter 3 or chapter 4 anywhere!?).

Therefore, despite the somewhat positive scores, I am recommending that this submission be rejected.

[1] Stolcke, "An Efficient Probabilistic Context-Free Parsing Algorithm that Computes Prexix Probabilities", 1992. https://arxiv.org/abs/cmp-lg/9411029

**Additional Comments On Reviewer Discussion:**

While some reviewers appreciated the study of Transformers' ability to learn CFGs, I broadly agree with the weaknesses outlined by Reviewer ura6, which was not adequately addressed by the rebuttal in my opinion.

---

### Decision · Program_Chairs · 2025-01-22

Reject